# Effect of Wheat Flour Integration with Blueberry Fruits on Rheological, Quality, Antioxidant, and Sensory Attributes of ‘French’ Bread

**DOI:** 10.3390/foods14071189

**Published:** 2025-03-28

**Authors:** Otilia Cristina Murariu, Gianluca Caruso, Gabriela Frunză, Florin Daniel Lipșa, Eugen Ulea, Alessio Vincenzo Tallarita, Anca Calistru, Gerard Jităreanu

**Affiliations:** 1Food Technologies, “Ion Ionescu de la Brad” Iasi University of Life Sciences, 3 M. Sadoveanu Alley, 700490 Iasi, Romania; otilia.murariu@iuls.ro (O.C.M.); gabriela.frunza@iuls.ro (G.F.); florin.lipsa@iuls.ro (F.D.L.); 2Department of Agricultural Sciences, University of Naples Federico II, Via Università, 100, 80055 Portici, Italy; gcaruso@unina.it (G.C.); alessiovincenzo.tallarita@unina.it (A.V.T.); 3Department of Plant Science, ‘Ion Ionescu de la Brad’ Iasi University of Life Sciences, 700490 Iasi, Romania; 4Department of Pedotechnics, ‘Ion Ionescu de la Brad’ Iasi University of Life Sciences, 700490 Iasi, Romania; gerard.jitareanu@iuls.ro

**Keywords:** *Vaccinium myrtillus* L., innovative bakery products, rheological properties, porosity, color, anthocyanins, antioxidant activity

## Abstract

Increasing interest is being devoted to innovative food products enriched with fruits and vegetables to enhance the nutritional and bioactive properties from the perspective of sustainable management. The addition (10, 15, and 20%) of blueberry fruits derived from two spontaneous flora varieties from the Rarău (G) and Ciocănești (C) mountains (Romania) into ‘French’ bread resulted in increased maximum breaking strength and mechanical work in spherical dough up to 10 and 15% in variety G, and deformation strength up to 20% integration; the untreated control displayed the highest values in the strips of dough. The 20% incorporation of both blueberry varieties in bread enhanced total, open, and closed porosity, maximum strength, gummosity, and chewiness, as well as titratable acidity, total soluble solids, vitamin C, flavonoids, anthocyanins, and antioxidant activity. Resilience and pH showed the highest levels in the untreated bread, which also exhibited the highest values of the color components ‘L’, ‘a’, and ‘b’ in both the bread crust and crumb. The untreated control showed the highest scores for some sensory features, and in most cases, an increasing trend with the fruit integration rising from 10 to 20% was recorded. The addition of blueberries represents an interesting strategy for creating bread as an innovative functional food under sustainable supply chain management.

## 1. Introduction

The bakery sector has a crucial impact on the food industry, considering that bread occupies a special place in most menus of different culinary cultures. Over time, the mentioned industry has experienced an increasing trend as a segment of global development and innovation [1]. In the latter respect, discovering valuable and viable solutions for creating functional flour-based products rich in naturally sourced nutrients beneficial to human health, such as antioxidants, vitamins, minerals, and fiber, is a major target of bakery manufacturers [2,3]. Among the most popular flour-based foods, there are bread, flour, pasta, biscuits, cakes, pizza, pastries, muffins, and buns [1,4], representing the most common foods for all consumer age categories and economic groups. Nowadays, both innovative fresh local products and those enriched with fruits and vegetables have been drawing increasing interest due to their high content of bioactive ingredients [5,6], as well as essential nutrients [1], such as lipids, complex carbohydrates like starch [4], minerals, and vitamins B [1]; however, they are poor in fiber, proteins [4], and antioxidants. In addition to the nutritional benefits, it is challenging to start with dough that has good rheological properties and to create baked products with appreciable sensory properties from gluten-free composite flours rich in fiber and proteins [4], even when adding fruit acids to meet consumer health benefit expectations [7].

Currently, researchers have been focusing on the development of innovative bakery products, adding various herbal ingredients and extracts in diverse combinations of elements [8]. These include essential oils, tannin substances, enzymes, organic acids, vitamins, pigments, minerals, anthocyanin-rich fruits, and grape pomace-infused flour, which enhance the dietary fiber content while also positively impacting the sensory attributes [9].

Another direction of research is oriented toward increasing the palatability of bakery products and their commercial value through the integration of ingredients such as extracts from rosemary, oregano, and lemon plants, which have a high preservative effect due to strong antioxidant and antimicrobial activity [10]. Raba et al. [11] proposed the development of bakery foods derived from the addition of lingonberry powder to spelt flour, considering that dried fruit powder integration provides nutritional health benefits to consumers of cereal products [3,12]. Also, there are expectations regarding the enhancement of physical, chemical, sensory, and microbial attributes of final products [3]. Fruits have an important effect on the quality of bread, pasta, and cakes in terms of elasticity, porosity, and volume because they can retain water and lipids. The mentioned fruit benefits are associated with their high content of dietary fiber, enhancing some physical attributes of bakery food [13,14]. Fruits have a high potential to improve the quality of dough, i.e., its rheological and technological properties, due to the high content of insoluble and soluble fibers being used as gelling agents, water binders, texture improvers, and fat substitutes [15]. The addition of blueberries can enhance bread texture due to the reduced number of alveoli and their circularity [16]; moreover, the gluten network protects the antioxidant compounds provided by the mentioned fruit, thus conferring healthier properties to the mentioned functional food. Blueberry integration also improves the color attractiveness and the overall sensory attributes of bakery products [1,17]. Research has shown that various berries, apples, oranges, mangoes, grape pomace, melons [3], peaches, strawberries, raspberries, and sour cherries [18] are frequently employed as fruit powders in the bakery industry.

Regarding the mentioned aspects, blueberries represent a high potential to be utilized as important bioactive ingredients in bakery products, derived from the species *Vaccinium myrtillus*. The latter is a shrub belonging to the Ericaceae family present in Europe, northern Asia, western Canada, and the United States; it is one of the most important berry species that have been established in Romania, both from spontaneous flora (mountain primroses) and commercial plantations that are being increasingly grown in agricultural areas [19]. It is a plant of ecological interest, not only for its fruits but also because it protects the forest soil from erosion and contributes to microbial enrichment [20] and humus formation [21]. Blueberry fruits are small berries (1–2.5 g) grouped in clusters, with purplish skin covered by a thick layer of blue bloom. Depending on the variety and climatic conditions, blueberries reach maturity from late July to September, presenting a staggered ripening [19]. These fruits are rich sources of antioxidant compounds [22] and contain: 80.0–84.6% water, 7.0–14.5% sugars, 0.56–1.13% acids (tartaric, malic, citric, benzoic, oxalic, succininic), 0.36–0.55% tannins, 0.5–0.6% pectic substances, 0.62–1.15% provitamin A, vitamins B1, B2, B3, C, PP, E, myrtilin, 31% oil in dried fruits, anthocyanins, and 0.23–0.32% mineral salts [19]. The consumption of blueberries, especially the spontaneously growing flora plant fruits (*Vaccinum myrtillus* L.—black blueberry, *V. uliginosum* L.—livid blueberry, and grown blueberry—*V. corymbosus* L.) is associated with the prevention of various cardiovascular and degenerative diseases [23], chronic inflammation, and intestinal cancer antiseptic due to their bactericidal substances, citric and benzoic acids, as well as the vasoprotective properties of anthocyanins, which confer beneficial effects to circulatory system. They are recommended after a heart attack, hemorrhages due to capillary fragility, and play a role in the regeneration of liver cells and hypoglycemia due to myrtillin lowering blood sugar [23]. In Romania, early-ripening blueberry varieties (Weymouth, Ivanhoe), mid-ripening (Bluecrop, Blueray), and late-ripening (Rubel, Pemberton, Coville) are grown [19].

Blueberries can be eaten fresh, dehydrated, or processed into juice, compote, syrup, liqueur, jam, jelly, marmalade, and cranberry; they also freeze well and are used in cakes, creams, mousses, and other confectionery products. Blueberry juice is used in winemaking to enhance color and increase acidity; however, these fruits cannot be used alone to make cider due to their benzoic acid content, which stops fermentation [19].

Along with the development of production and assortment of bakery products, it is necessary to create innovative foods that enrich bread with antioxidant-based ingredients, thus offering solutions to satisfy the demands of consumers for beneficial foods from the perspectives of health and environmental safety. Referring to the high consumption of bakery products and the extremely important diversity resulting from the suitability of this product to be combined with various fruits, it is interesting to identify new variants of bread integrated with blueberry pulp. Therefore, the main goal of the current research was to improve the nutritional profile of bread by incorporating blueberry pulp and evaluating the quality parameters, as well as consumer acceptability, for these innovative products.

In the present study, the ‘French’ bread was enriched with blueberry fruits grown in two different areas (Giumalău Mountains and Ciocănești Mountains) from two spontaneous flora blueberry varieties, in the form of crushed pulp at the concentrations of 10, 15, and 20%. The latter range of integration percentages was chosen based on both literature reports [11] regarding either the same fruit or similar ones and on the expected technical–economic balance related to the manufactured product. It was hypothesized that the diverse varieties and concentrations of blueberry fruit would change the textural properties of ‘French’ bread, significantly impacting bread porosity, quality, color, and antioxidant constituents.

## 2. Materials and Methods

### 2.1. Research Location and Experimental Protocol

Research was conducted at the Microsection of Bakery Products and Quality Analysis Laboratories of the Food Technologies Department (Iasi University of Life Sciences, Romania, 47°11′76′′ N, 27°33′71′′ E), in 2024, with the aim to assess the effects of adding blueberry fruit to ‘French’ bread on the rheological, quality and antioxidant characteristics, and mineral composition of the final product. The fruits of two varieties of spontaneous flora blueberry (*Vaccinium myrtillus* L.), both harvested in September 2024, were used: the first variety (named G) came from the Giumalău Mountains (Carpații Orientali, highest peak at 1858 m, 47°24′ N, 25°30′); the fruits of the second variety (named C) were collected in Ciocănești (Suceava, highest peak at 855 m, 47°28′51′′ N, 25°16′44′′).

The experimental protocol was based on the comparison between six treatments derived from the combination of three addition percentages (10, 15, and 20%) of blueberry fruits from the two mentioned varieties (G and C) in white bread, plus an untreated control, using a completely randomized design with three replicates. The latter procedure allowed for all possible comparisons between the six treatments and each of them with the untreated control. In this respect, we could observe and describe all the trends of each examined variable as a function of the addition percentage for each variety and the comparison between the two varieties at each addition percentage. The untreated control, based on the same basic ingredients as the experimental treatments except for the blueberry additions, serves as an essential reference for assessing the possible differences arising from *V. myrtillus* fruit integrations.

After harvesting (2000 g per variety), the samples were immediately frozen at −18 °C until November, when they were used for laboratory analyses.

### 2.2. Preparation of the Raw and Auxiliary Materials

The control sample was obtained by following a standardized recipe and technology, including a blend of the following raw materials: white flour (60%), water (35%), fresh yeast (3%), salt (1%), and sugar. The recipes of the control bread sample and the blueberry fruit-enriched breads are presented in Table 1, which outlines the proportion of each ingredient used in the production process.

The premium quality white wheat flour used in this research was purchased from a local retail market (000 extraction level). Flour is considered the heart and soul of the product, and the type of flour significantly impacts the quality of bread. The blueberry fruits were acquired frozen from a Romanian fruits and vegetables warehouse located in the Câmpulung Mountain area (S.C. Diligent Sadova SRL, manufacturer of ecological products in Sadova, Suceava County) and from a local retailer in Ciocănești, Suceava (Figure 1). The biologically active compressed yeast *Saccaromyces cerevisiae,* produced in 500 g blocks, was purchased from a local market. Afterward, the flour was stored at 21 °C, blueberries were kept frozen at −18 °C, and the fresh yeast was stored in a 700 L capacity refrigerator at 0–4 °C until sample preparation.

The preparation stages aimed to bring the raw and auxiliary materials to the optimal physical condition for the preparation of dough (Figure 2). Flour was sifted to remove coarse and metallic impurities accidentally present in it. Water preparation consisted of mixing cold water with water at 60 °C, and the water was brought to the appropriate temperature of 29–30 °C for a normal dough fermentation course.

The suspension of yeast was carried out in order to distribute it as evenly as possible in the dough mass, achieving homogeneous fermentation. The yeasts, through alcoholic fermentation, release carbon dioxide, which boosts dough growth, and the lactic bacteria create a certain level of acidity favorable for yeast development. During dough kneading and fermentation, as well as baking, an intensive multiplication of biologically active microorganisms is achieved and maintained. Salt can be used when dissolved in the form of a filtered solution. The blueberries were defrosted at 0–4 °C, then ground and homogenized using a Philips mixing robot HR 320/700, and finally mixed with water at 37–40 °C.

### 2.3. Bread Preparation

The technological flow related to ‘French’ bread production derived from *Triticum aestivum* flour (000 type), with the dough being prepared using the direct method, includes the steps reported in the chart-flow presented in Figure 3 and Figure 4, in compliance with the ISO 6820-1985 regarding ‘Wheat flour and rye flour-General guidance on the development of bread-making tests’ [24].

Seven batches of bread, corresponding to six formulations (B), including blueberry additions plus an untreated control, with three loaves per batch, were manufactured in the same ambient conditions according to the recipes shown in Table 1. The process included the following stages: Scaling, Mixing, Kneading and Folding, Fermentation, Shaping, Rising, Baking, and Cooling (Figure 3 and Figure 4).

All the ingredients were thoroughly mixed in two phases of speed, until a homogeneous batter formed using a 7SN Fimar spiral mixer^®^ (Fimar, Rimini, Italy) with a production capacity of 7 kg. The first phase was slower, lasting up to about five minutes, during which the flour was hydrated with water and all the ingredients were incorporated into a fluid mass. Then, the dough was kneaded for an additional five minutes at a higher speed, and meanwhile, the development of the gluten began. At the end of the latter phase of kneading and folding, the development process started, and then, the initial growth took place, due to the presence of yeasts, which began to consume the sugars and starch, transforming them into gases, which filled the dough with air for 60 min at 29 °C and 85% humidity (Figure 4). Subsequently, the dough was equally divided into 3 portions, conferring a consistent size and distribution to the samples (950 g each). The latter were immediately used to determine textural properties, and to prevent fermentation, each was divided into two portions as follows: one half was divided into three parts of 50 g and modeled as 100 × 100 mm spheres, which were spread on the flat with 5 mm thickness, and the second half was modeled as spheres spread into a flat form and then divided into strips, 100 mm long, 15 mm wide, and 5 mm thick.

The 3 portions of each experimental treatment were shaped in a final form and structure for the dough, which was formed into a tray loaf. The final fermentation was carried out at 29 °C for 50 min, and then the samples were baked after sprinkling and inking at 200 °C for 50 min in a Zucchelli Forni electrical rotary convection oven (Zucchelli Forni, San Pierino, Italy). In the first 15 min, the steam was used to prevent the crust formation, which will determine the blocking sample expansion in the oven. After baking, the bread samples were cooled down at 20 °C for 3 h, then were packed in polyethylene film to preserve freshness and kept at 20 °C in the first 24 h and at 2–4 °C for the next 9 days. One day after preparation, the sensory analysis was performed.

### 2.4. Determination of Bread Porosity and Rheological Properties

To investigate the internal structure of the bread samples, they were scanned using the Computed Tomography method, which is non-destructive and non-invasive, by a micro-CT SkyScan 1273 (Bruker (Billerica, MA, USA)). All the samples were scanned using the same protocol, at a voxel size of 30 µm (micrometers), with no special filter. NRecon software (version 2.0) was used for the reconstruction of the images, and CTAn software (version 1.18) for the porosity analysis, i.e., open, closed, and total porosity, expressed in % (both software programs by Bruker).

The textural properties (maximum breaking and deformation strength of dough/bread, firmness, elasticity, resilience, gumminess, chewiness, and cohesiveness of bread) were assessed in 3 replicates by a food texture analyzer (Mark 10, New York, NY, USA) on samples consisting of three 25 mm thick slices of bread placed on the plane surface of the texturometer base.

Maximum breaking strength, representing the maximum resistance of the dough/bread to breaking, and maximum deformation force, representing the maximum deformation of the dough/bread, refer to the extensibility of bread and are determined through uni- and bidimensional elasticity expressed in Newtons (N). Dough extensibility is an important indicator of bread dough quality, and its viscoelastic properties are essential. A dough that is too difficult to handle will often result in a deformed final product that does not maintain the desired structure. The determination of these characteristics provides a good criterion for how dough will behave during the baking process. Elasticity was measured by a “V”-shaped bread press positioned at an angle of 60° to reproduce the compression and breaking stress, providing a standardized and repeatable version of the consumer’s freshness test; the lower the force, the softer the product’s perception, determining the elasticity based on how easily the product recovers from compressive stress. To measure the unidimensional elasticity, dough was introduced into a laminating machine from which 5 mm-thick sheets were extracted, and dough strips with dimensions of 100 × 10 × 0.5 mm were cut from the sheets. The dynamometer, together with the tension probe (type L), was raised above the lower disk and then lowered under the lower disk of the texturometer. The dough strips were positioned centrally on the texturometer disk, then secured with a lid. After setting the texturometer program, the dynamometer was commanded to lift together with the tensile probe at a constant speed until it stretched the sample and broke it.

The bidimensional elasticity is determined using the previously mentioned technique associated with the specifications of 5 mm-thick sheets. Dough sheets with 100 × 100 × 100 mm dimensions were cut. The dynamometer, together with the spherical head probe, was lowered under the lower disk of the texturometer. After setting the texturometer program, the dynamometer is commanded to lift together with the spherical-headed probe that stretches the sample and breaks it.

Both in the case of unidimensional and bidimensional elasticity, the values of force (N) and displacement (mm) were recorded and can be reproduced as a tensile curve in the same coordinates mentioned. The interpretation of the recorded values was carried out using the GraphPad Prism^®^ (GraphPad Software (version 10.1.2), Boston, MA, USA) program.

The elasto-plastic constant was calculated with the formula k = Le/Lp, where Le is the plastic mechanical work (plastic strain energy) [mJ], Lp is the elastic mechanical work (elastic deformation energy) [mJ] [25].

Firmness was measured by a cylindrical probe with a 36 mm diameter pressing the surface of the bread sample at a 200 mm min^−1^ speed and expressed in N.

The determinations mentioned were repeated one week after manufacturing *Vaccinium myrtillus,* adding bread, and, as no significant differences arose compared to the initial values, the results have not been reported.

### 2.5. Determination of Quality and Color Parameters of Bread

The chemical reagents (NaOH, AgNO_3_, K_2_CrO_4_, DPPH, Folin–Ciocalteu’s reagent, gallic acid, and solvents) were of analytical grade and purchased from Sigma Aldrich Chemie GmbH (Munich, Germany).

The pH and titratable acidity were measured in each sample as follows: 25 g of bread was blended with 50 mL of distilled water, which was further added upon homogenization to bring the volume up to 250 cm^3^, leaving the latter to rest for 15 min to decant. Fifty ml were taken from the clarified part to measure the pH, after which five drops of 2% phenolphthalein alcohol solution were added and titrated with 0.1 N NaOH until the pink color remained stable for 1 min. The results were expressed as means of three repetitions in mL 0.1 N NaOH [26].

The moisture content (%) of bread crumbs was measured according to SR 91/2007 [26], by placing five g of sample in a dryer at 130 °C for 1 h. Proteins with exclusive functional properties (gliadin and glutenin in wheat flour) for breadmaking were analyzed using the Kjeldahl method in compliance with AOAC International [27], based on catalyzed digestion with sulfuric acid. The total organic nitrogen was converted to ammonium sulfate. The digest was neutralized with alkali and distilled into a boric acid solution. The borate anion forms were titrated with standardized acid, converted to nitrogen, and the final result represents the bread proteins. Lipids were determined using the Soxhlet method [27]. For semicontinuous extraction of solvent, the latter builds up in the appropriate chamber for 5–10 min and completely surrounds the sample before siphoning back to the boiling flask. Lipid content was measured by the sample weight loss.

Mineral substances (ash) were determined as the residues resulting from the calcination of the analyzed sample at 550 °C for 6 h [27], expressed in %. The NaCl content was measured upon titration of chlorine ions with AgNO_3_ in the presence of K_2_CrO_4_ as an indicator and was expressed as the mean of three replicates in mL 0.1 N AgNO_3_ [27].

The color components of bread were determined in three replicates by a Minolta Colorimeter-400 trichromatic reflectance colorimeter with Spectra Magic NX 1.3 software (Konica Minolta Sensing INC., Osaka, Japan), according to the CIE Lab system (L*, a*, b*) at ambient temperature [1].

The mentioned determinations were repeated one week after the production of blueberry-added bread stored at 2–4 °C, and as no significant differences arose compared to the initial values, the results have not been reported.

### 2.6. Evaluation of the Phytochemical Profile

The phytochemical profile was evaluated by determining the content of vitamin C, total polyphenols, total monomeric anthocyanins, total flavonoids, and antioxidant activity of the sample extracts.

#### 2.6.1. Extraction of Biologically Active Compounds

The biologically active compounds were extracted from bread by an ultrasound-assisted method. One gram of each bread sample was homogenized, in the first stage, with 9 mL of solvent (70% ethanol) and then with 1 mL of 1% citric acid solution at room temperature for 5 min. The samples were then subjected to an ultrasound treatment for 40 min at a 40 kHz frequency and a temperature of ≤30 °C. The resulting recovered crude extract was centrifuged for 10 min at 5000 rpm and 4 °C. After separation, the supernatant was collected, and the following antioxidants were analyzed: vitamin C, total flavonoids, polyphenols, anthocyanins, and antioxidant activity using the DPPH method.

#### 2.6.2. Determination of Vitamin C

Ascorbic acid has the property of reducing Tillman’s reagent to the corresponding leucoderivative. The principle of the method consists of extracting ascorbic acid from the sample with a 2% oxalic acid solution (sample: 2% oxalic acid-1:5), and then 5 mL of the extract was titrated with Tillman’s reagent until a light pink color appeared. The ascorbic acid content was expressed as mg 100 g^−1^ f.w. [27].

#### 2.6.3. Determination of Total Polyphenol Content

The total polyphenol content was measured as described by the Folin–Ciocalteu method [28]. A volume of 0.2 mL of extract was mixed with 15.8 mL of distilled water and 1 mL of Folin–Ciocalteu reagent. The mixture was homogenized and kept at rest for 10 min; then, 3 mL of 20% sodium carbonate solution was added. Next, the mixture was thermostated for 60 min at room temperature in the dark, after which the absorbance at a 765 nm wavelength was analyzed spectrophotometrically against a control sample. The results were expressed in mg of gallic acid equivalents per gram of dry weight (mg GAE g^−1^ d.w.) using the gallic acid standard curve (y = 1.6991x − 0.0256; R^2^ = 0.9837).

#### 2.6.4. Determination of Flavonoid Content

The determination of total flavonoid (TFC) content was performed using the spectrophotometric method based on the reaction between aluminum chloride and phenolic compounds, as described by Turturică et al. [28]. A volume of 0.25 mL of extract was mixed with 1.25 mL of distilled water and 0.075 mL of a 5% sodium nitrite solution. The mixture was kept at rest for 5 min, after which 0.15 mL of 10% aluminum chloride solution was added. After a further 6 min rest, 0.50 mL of 1 M sodium hydroxide solution was added, and the absorbance of the mixture was evaluated spectrophotometrically at a wavelength of 510 nm, compared to a control sample (without extract). TFC was determined based on a standard curve with catechin (y = 2.8919x + 0.006; R^2^ = 0.9968), and the results were expressed in milligrams of catechin equivalents g^−1^ d.m. (mg EC g^−1^ d.w.).

#### 2.6.5. Determination of Total Monomeric Anthocyanin Content

The content of total monomeric anthocyanins was measured as reported by Turturica [28] and expressed as milligrams of cyanidin-3-glucoside (C3G) equivalents per gram of dry matter (mg C3G g^−1^ d.m.). The absorbance was determined spectrophotometrically at wavelengths of 520 nm and 700 nm using different buffer solutions, namely potassium chloride (0.025 M) at pH 1.00 and sodium acetate (0.40 m) at pH 4.50. The concentration of total monomeric anthocyanins was determined based on the Equation (1):
Total monomeric anthocyanins = (A × MW × DF)/(ε × L)(1)
where A = [(A520–A700) pH 1.00 − (A520–A700) pH 4.50]; MW—molecular mass, MW = 449.2 g mol L^−1^ C3G; DF—the dilution factor; ε = 26900, molar extinction coefficient for C3G; L—width of the cuvette (1 cm).

#### 2.6.6. Determination of Antioxidant Activity by the DPPH-2,2-Diphenyl-1-Picrylhydrazyl Method

The degree of neutralization of the free radical DPPH (2,2-Diphenyl-1-Picrylhydrazyl) was performed as reported by Turturică et al. [28], with some adaptations. A volume of 0.10 mL of the extract was mixed with 3.90 mL of methanolic DPPH solution (1.50 10^−4^ M). The mixture was homogenized and then kept for 30 min at 25 °C in the dark. The absorbance of the reaction mixture was then read spectrophotometrically at a wavelength of 515 nm. The blank was prepared by adding 0.20 mL of methanol instead of the extract. The results were expressed as TE g^−1^ d.m. and as percent inhibition using Equation (2):I (%) = (Ablank − Asample)/Ablank × 100(2)
where Ablank = absorbance value of the blank; Asample = absorbance value of the sample.

The calibration curve using Trolox is represented by the equation y = 0.45x + 0.0075, R^2^ = 0.993.

The mentioned determinations were repeated one week after producing *Vaccinium myrtillus*-added bread stored at 2–4 °C, and as no significant differences arose compared to the initial values, the results have not been reported.

### 2.7. Sensory Evaluation of Bread Products

The sensory parameters were evaluated through a panel test, in which the hedonic scale was used for panelist preferences and the Quantitative Descriptive Analysis test for a detailed sensory profile relevant to general aspects, color, texture, as well as taste and smell-related attributes of bread samples according to the specifications of ISO 4121:2002 [29]. In this respect, the selection of 20 individuals was considered sustainable and reliable, according to the specifications of ISO 8586:2023 (for panelist selection), in terms of our activity to instruct and train them in sensory evaluation [30]. The panelists underwent training in 4 sessions, during which the criteria for evaluating each of the bread attributes examined were explained to them, using standardized references for general aspects, color, texture, as well as taste and smell-related attributes. Moreover, training was conducted on a standardized intensity scale, benefiting from the instructions given by some professors included in the author list and by experts in this field for the descriptive evaluation. They were aged between 23 and 50 (10 females and 10 males), non-smokers, without known cases of food allergies, regular consumers of bread products from the Iasi University of Life Sciences, Department of Food Technology. They were requested to provide their ratings based on a 9-point scale, referring to the following attributes: general aspect, crust thickness and color, core color, aeration, humidity, volume, texture and structure, elasticity, spreadability (by friction), flexibility (by bending), and the main aromas felt (smell + taste) in the samples [31], such as acidity, saltiness, sweetness, yeast, grain, citric, minerals, fruits, and berries. For all parameters under evaluation, 0 indicates extreme dislike, and the maximum value (9) represents the highest level of liking. The sensory analysis of bread was organized following laboratory ethical guidelines, and the consents of each evaluator were signed in compliance with the European Union Guidelines of Ethics and Food-Related Research (protocol no. 19.290/2024) [32]. The samples were prepared in advance by slicing them uniformly, respecting the standardized dimensions for each slice (5 × 5 cm), coding them randomly with a 3-digit number, and serving them randomly in a Sensory Analysis Laboratory corresponding to the specific conditions for organizing the tasting sessions. The tests were performed inside sensory booths at an ambient temperature of 20–22°C, in neutral light conditions, without foreign smells. During the process of sensory evaluation, the palates of the panelists were neutralized with water and salt crackers. The sensory evaluation related to each experimental treatment was repeated thrice at 24 h intervals.

The mentioned determinations were repeated one week after manufacturing blueberry-added bread stored at 2–4 °C, and as no significant differences arose compared to the initial values, the results have not been reported.

### 2.8. Statistical Analysis

Data were processed by analysis of variance (ANOVA), and mean separations were performed through Duncan’s test, with reference to *p* < 0.05 in cases of statistical significance of a measured variable, to simultaneously compare the experimental treatments with each other and with the untreated control. Moreover, a principal component analysis (PCA) was carried out. SPSS software version 29 was used for data statistical processing.

## 3. Results and Discussion

### 3.1. Rheological and Quality Characteristics of Bread

As can be observed in Table 2 regarding the parameters measured on flat spherical dough, the maximum breaking strength was recorded with the 10 and 15% variety G blueberry addition, not significantly different from the 20% addition and untreated control; the latter showed the highest value referring to dough flat samples, followed by the aforementioned 10 and 15% blueberry treatments. The maximum deformation of spherical dough samples was shown by the 20% var. G blueberry addition, which was not significantly different from the untreated control; the latter displayed the highest value of the mentioned parameter in the dough flat samples, while the 10% blueberry var. C displayed the lowest. The highest value of mechanical work was recorded under the 10% blueberry var. G addition, which was not significantly different from the 15% addition; the untreated control attained the highest level of mechanical work, and the 20% blueberry var. C addition had the lowest (Table 2).

Bread porosity generally increased with the rising addition of blueberries; however, only the highest percentage of cv. C fruits led to higher values of total and open porosity compared to the untreated control, which showed lower closed porosity only when compared to the 20% cv. G fruit integration (Table 3). A similar trend upon the augmentation of blueberry fruit percentage was recorded for maximum strength, which was not significantly different from the untreated control (Table 3), gummosity, and chewiness (Table 4), as well as titratable acidity, total soluble solids (T.S.S.), and lipids (Table 5), vitamin C, flavonoids, anthocyanins, and antioxidant activity (Table 6). Different trends were recorded for firmness and resilience, showing the highest level in the untreated bread (Table 4). This was not significantly different from the lowest blueberry additions for firmness and from the highest incorporation of var. G fruits for resilience; pH also displayed the highest value in the untreated control (Table 5). The 10% addition of var. G blueberries optimally fostered the mineral substance content compared to the untreated control, and NaCl, in relation to the three addition percentages of var. C blueberries. Elasticity and cohesiveness (Table 4), as well as proteins (Table 5), were not significantly affected by the integration of blueberries into the bread.

The main effects of the two experimental factors applied on the three color components of bread are presented in Table 6. The interaction between blueberry variety, fruit addition percentage, and production stage was significant on the three color components of bread (Figure 5). In this respect, the untreated control showed the highest values of the color component ‘L’ in the dough, bread crust, and crumb; upon the blueberry addition, the bread crust attained the highest L levels while the crumb had the lowest ones (Figure 5); as the blueberry addition increased, the ‘L’ component generally decreased, except in the bread crumb with the variety C. In the untreated control, the color component ‘a’ was the lowest in the dough and bread crumb and the highest in the crust. Upon the addition of blueberries, the bread crumb generally showed the lowest values of ‘a’, whereas the comparisons between the dough and crust were controversial. The untreated control showed the highest values of the color component ‘b’ in the dough, crust, and crumb; with the addition of blueberries, the highest levels of ‘b’ were recorded for the crust and the lowest for the dough. Increasing the percentage of blueberry addition led to controversial trends in the three stages of bread production (Figure 5).

Poiana et al. [33] recorded the maximum values of elasticity and porosity in bakery products in the untreated control, with decreases in elasticity by 5.9 and 19.9%, and in porosity of 7.8 and 9.0%, in cakes and rolls, respectively, under the 25% grape pomace addition. Similar reports of Plustea et al. [34] indicate a significant decrease in elasticity and porosity of bread with the addition of 10–30% lupin flour, and of Dossa et al. [35] regarding the inclusion of baobab pulp flour into wheat flour. The two rheological parameters mentioned are influenced by both the dough manufacturing procedure, particularly gluten content and ability to retain fermentation gases, as well as the amount and structure of hydrolysable starch. [34]. In the Poiana et al. [33] experiment, the increasing addition of grape pomace elicited a rise in the spread ratio (SR) of biscuits, as also reported by other authors upon the addition of non-cereal flours to dough, presumably due to the decrease in gluten network and an augmentation of fiber [36].

At the initial mixing, the ingredients are evenly distributed, the spherical protein particles break down, and the flour is hydrated, which triggers the creation of a three-dimensional viscoelastic structure capable of retaining gases, with polymeric proteins as the main component [37]. Under heating, starch granules swell upon absorbing the water available in the medium, and the amylose chains increase viscosity due to their leaching out into the aqueous intergranular phase. The latter phenomenon continues until the reduction in mechanical shear stress and temperature causes physical degradation of particles, decreasing viscosity [37]. Raba et al. [11] recorded a texture decrease due to amylolytic activity upon 5% to 15% lingonberry addition; moreover, the 10–15% lingonberry fruit powder integration into spelt flour resulted in the optimal bakery products, considering that exceeding the mentioned percentages caused the dough to become less stable and harder to process, which consequently lowered the volume and appearance attractiveness of the final products.

In previous research, significant moisture reduction was recorded in pastry products upon increased addition of powder from grape [33,36,38] and lingonberry [11], but moisture augmentation followed the integration of fruit [39,40]. Increasing trends were found in terms of mineral and protein content in bakery products under the fortification with fruit [11,33,41,42] or mushroom powder [43]. However, Nakov et al. [36] reported the opposite protein trend. The mentioned impact of the vegetable powder additions on the quality characteristics of the bakery product is associated with the composition of the blended ingredients used.

### 3.2. Antioxidant Compounds and Activity of Blueberry Fruit-Added Bread

The rising addition of 10 to 20% of blueberry fruits of both varieties to bread generally led to increased values of vitamin C, total polyphenols, flavonoids, and anthocyanins (Table 7), as well as antioxidant activity (Figure 6); the untreated control always showed the lowest values.

Phenolic compounds are the widest class of phytochemicals, providing protection to plants against abiotic and biotic stresses [44,45], and include different compound types, among which flavonoids and anthocyanins, with several biological effects [46]. In previous research [33], a 7.6-fold higher antioxidant activity was recorded in pastries fortified with 25% grape powder integration, compared to the untreated control, with decreasing values from rolls to cakes. In other research, the enhancement of antioxidant activity was recorded in bakery products upon the increasing ingredient addition of grape from 5 to 20% [47], from 20 to 30% [38], and from 10 to 30% [48].

Poiana et al. [33] reported the highest total flavonoid content at a 25% grape powder integration level, which was 8-fold higher than that of the untreated control, while Maner et al. [48] reported similar results at a 10% grape integration.

According to Gaita et al. [49] and to Seczyk et al. [50], phenolic compounds bound to food matrix components contribute to limiting the enhancement of the antioxidant properties, and their remarkable presence in the integrated materials can foster the bioavailability of flour-based food chemical compounds. Particularly, the interactions between phenolic species and proteins in the matrix lower the fortification efficiency [50].

Poiana et al. [33] found significantly higher DPPH values in pastry products with grape powder addition, compared to the control, presumably due to the increase in total phenolic compounds in the added ingredient [51,52].

Other authors recorded a DPPH increase by 17.5-fold on average, compared to the untreated control, upon the addition of 25% grape [33,42,50], due to the high contribution of total phenolics [53]. In another investigation [11], 113, 63, and 68% higher DPPH inhibition than the untreated control was recorded in cookies, muffins, and brioches, respectively, with the addition of 25% lingonberry powder to spelt flour, due to the abundance of flavonoids, polyphenols, and vitamins in the added fruits. In another research related to bread processing [34], the DPPH inhibition values ranged from 35 to 55% in wheat composite flour added with the addition of 10 to 30% lupin. Sturza et al. [54] found an increase in the percentage of inhibited DPPH free radicals in pastry products, reaching 74% with 4% sea buckthorn integrated into gingerbread and up to 18% with the same percentage of *Hippophae rhamnoides* included in sponge cake. Other authors reported that the antioxidant activity increased up to 20 with the addition of 5% sea buckthorn flour [55], and a DPPH value of 89 in *H. rhamnoides* powder-integrated biscuits [56]. Raba et al. [11] reported increasing trends of total antioxidant activity and flavonoids in fortified pastries, up to six times higher than the untreated control with the addition of 25% lingonberry powder, with the top values in cookies, followed by muffins and brioches. Increasing total polyphenolic content in cookies was also found in other studies: upon the integration with pomegranate rind powder up to 39% [41]; through the fortification with 7.5% date seed powder [57]; with the addition of mahaleb seed powder to wheat flour [58]; and through the fortification with mushroom powder [43]. In previous research [59], the 15% addition of pumpkin peel, flesh, and seed powders to white wheat flour elicited the top flavonoid content in biscuits (61 mg CE 100 g^−1^), and in flakes (373 mg RE kg^−1^) and muffins (532 mg RE kg^−1^) from spelt flour [60].

### 3.3. Sensory Features of Blueberry Fruit Added Bread

The general aspect of bread (Table 8) attained the highest value in the untreated control and with the 15% addition of variety C blueberry, with the mentioned percentage being the most effective also for variety G. The crust thickness was highest at the 10% blueberry addition, only in var. C, which did not significantly differ from the untreated control. The highest values of crust color were recorded with the 20% blueberry addition for both varieties, significantly higher than those associated with the untreated control (Table 8). The untreated control and the 20% var. G blueberry addition showed the highest score for core color, whereas the core aeration score of the control and the 20% var. C blueberry integration was higher only compared to the 10% var. G addition.

Regarding the sensory attributes shown in Figure 7 and Figure 8, the untreated control displayed the highest value of grain; a higher level of elasticity than the highest blueberry percentages; generally higher values of sweetness compared to the lowest blueberry additions; the lowest spreadability, citric, mineral, fruit, and berry taste; no significant difference from the blueberry addition treatments generally regarding core texture and structure, flexibility, and salt taste; and a lower acid taste than the 10% variety C blueberry addition.

As for the comparison between the blueberry addition treatments, in most cases (core texture and structure, flexibility, sweetness, citric, fruit, and berry taste), a general increasing trend with the integration rising from 10 to 20% was recorded. The opposite trend was shown by elasticity, acid, salt, and grain taste. The trends of mineral taste were controversial.

The overall effects of the experimental treatments on the sensory attributes have been clustered in a heatmap (Figure 9).

The sensory analysis was performed to determine consumer acceptability. In previous research [43], cookies integrated with mushroom powder showed better overall sensory features than the control under the *P. ostreatus* addition but performed worse when blended with *A. bisporus*. Poiana et al. [33] recorded the best taste, flavor, and overall acceptability of both biscuits and rolls with the 10% grape pomace addition, similar to the findings of Lou et al. [61] and Boff et al. [62]. In other research [48], the highest scores of sensory attributes, i.e., texture, mouth sensation, aroma, taste, and overall acceptability, were recorded in cookies added with up to 15% grape pomace. Nakov et al. [36] revealed the best nutritional value and sensory characteristics in cakes with a low grape powder addition percentage (4%), whereas the 6% integration led to the best texture.

The mentioned studies reported a decreasing trend of appearance appreciation with over 10% grape ingredient addition due to the sample’s darker color. In this respect, the use of 10% grape powder as a natural substance to color bakery food showed a significant influence on the product’s browned appearance [62,63], as well as mushroom integration resulting in lower cookie whiteness [43].

According to previous reports [50,64], the inclusion of *Vaccinium vitis-idaea* fruit powder in pastry products led to the best scores of taste/chewing, aroma, texture/porosity and overall acceptance. In particular, Raba et al. [11] recorded the best scores in 10% lingonberry-added muffins, whereas the 25% percentage had the worst impact.

### 3.4. Principal Component Analysis

The two principal components presented in Figure 10 accounted for 75.3% of the total variability of the examined variables. Most of the analyzed parameters of the bread, such as rheological, quality and antioxidant properties, were best affected by the 15 and 20% additions of both varieties of blueberries to bread. The untreated control mainly displayed the highest values of dough and bread color components L and b. The lowest blueberry integration percentages were best associated with the dough rheological characteristics and bread protein content (Figure 10).

Particularly, the untreated control is positioned in the negative extreme of the *x*-axis, indicating that most of the variability of the dependent variables was due to the blueberry-enriched treatments, which showed a positive direction along the *x*-axis with increased blueberry addition.

The experimental variables occupying the positive side of the *x*-axis represent the bread characteristics better affected by blueberry fruit additions (i.e., higher antioxidants, vitamin C, flavonoids, anthocyanins, soluble solids, chewiness); in contrast, other bread characteristics, such as deformation strength, protein content, cohesiveness, and crust, were more directed toward the untreated control.

Some parameters, such as porosity, acidity, and lipids, tend toward the treatments associated with the highest blueberry additions occupying the positive part of the *y*-axis, in contrast to proteins and mineral substances.

## 4. Conclusions

From the present research, it emerged that the use of *Vaccinium myrtillus* fruits derived from Romanian spontaneous flora varieties, in the range of 10 to 20%, to fortify ‘French bread elicited the enhancement of some rheological, quality, and sensory attributes, particularly referring to the highest blueberry addition. Interestingly, the latter integration generally led to improved dough textural characteristics, as well as the highest values of bread bioactive compounds, such as vitamin C, flavonoids, anthocyanins, and antioxidant activity. However, the highest fruit addition negatively affected the color components and some sensory attributes, such as grain taste and elasticity perception of the bread. Based on the outcome of this study, interesting perspectives relate to the creation of innovative functional food derived from ‘French’ bread manufacture, combining wheat flour with blueberry fruits, with the aim of enhancing the overall product properties under a sustainable strategy. In this respect, the higher production cost of blueberry-added bread (1.06, 1.15, and 1.30 euros associated with 10, 15, and 20% additions, respectively, compared to 0.50 euros for the untreated bread), is justified by the healthier properties of the aforementioned bioactive compound-enriched food.

Future research should explore the following topics relevant to bread production: (i) long-term stability of antioxidants during bread storage; (ii) using different fruit-processing methods, such as freeze-drying, in comparison with fresh pulp for fortification; (iii) investigating consumer acceptability under actual market conditions.

## Figures and Tables

**Figure 1 foods-14-01189-f001:**
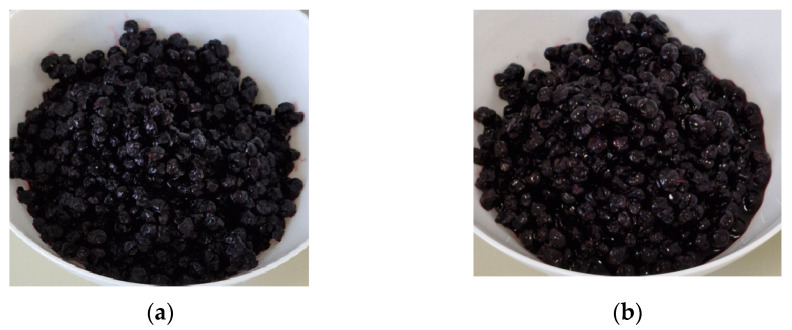
(**a**) Spontaneous blueberries harvested from Giumalău Mountain. (**b**) Spontaneous blueberries harvested from Ciocănești Mountain.

**Figure 2 foods-14-01189-f002:**
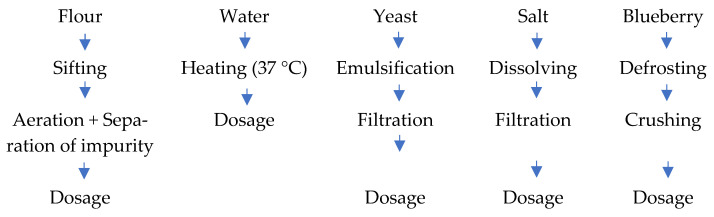
The preparation stages of raw and auxiliary materials.

**Figure 3 foods-14-01189-f003:**
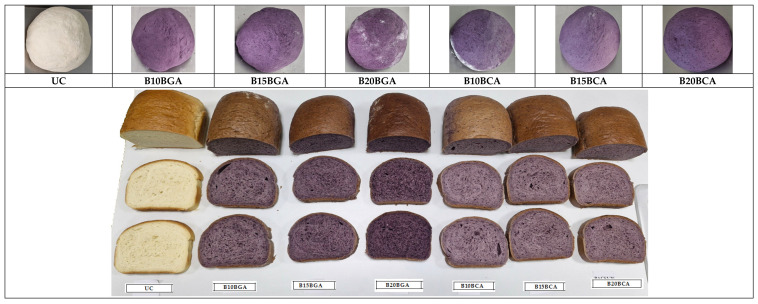
The aspect of the dough and developed bread samples.

**Figure 4 foods-14-01189-f004:**
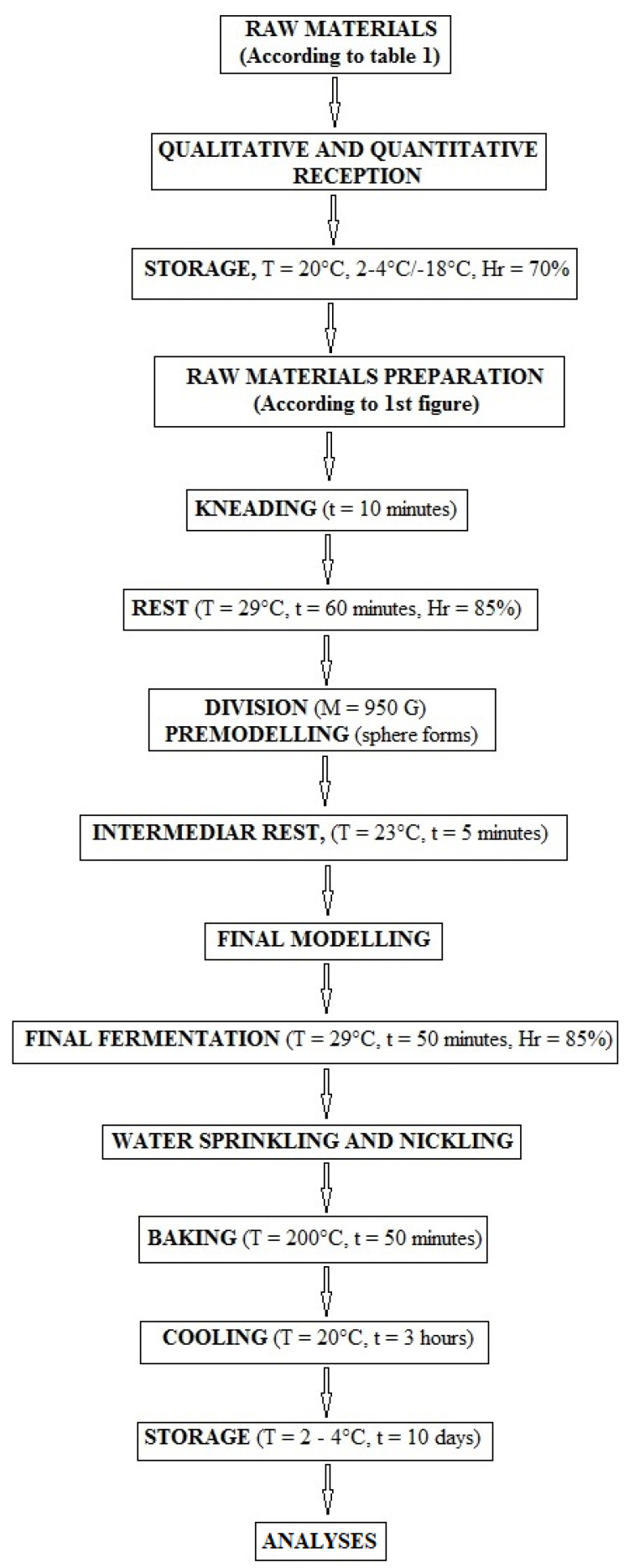
Technological scheme for functional bread preparation.

**Figure 5 foods-14-01189-f005:**
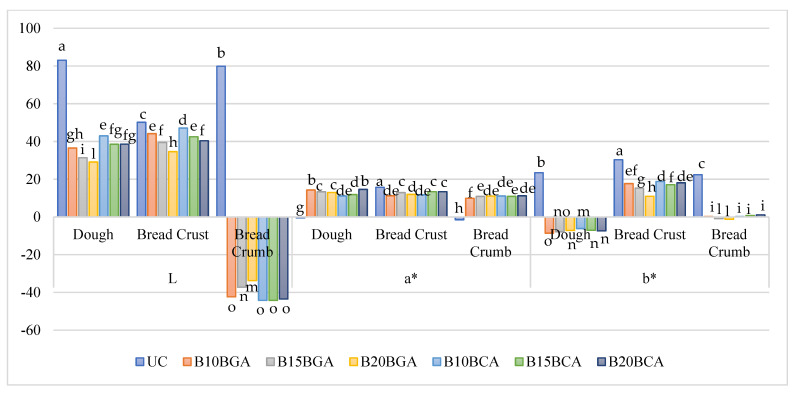
Interaction between blueberry variety, fruit addition percentage, and production stage on bread color components. Values followed by different letters are significantly different at *p* ≤ 0.05 according to Duncan’s test.

**Figure 6 foods-14-01189-f006:**
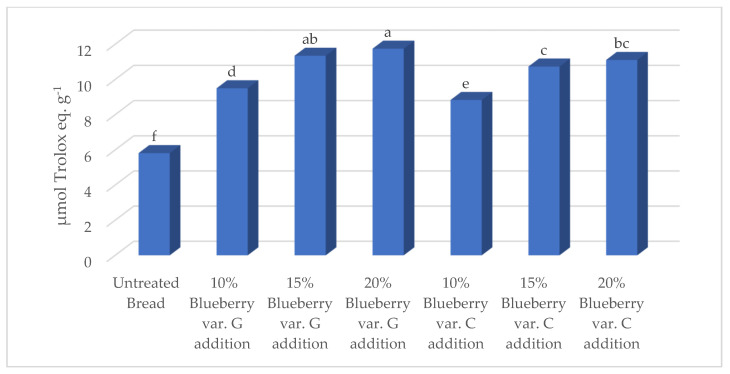
Effects of blueberry variety and fruit addition percentage on DPPH antioxidant activity. Values followed by different letters are significantly different at *p* ≤ 0.05 according to Duncan’s test.

**Figure 7 foods-14-01189-f007:**
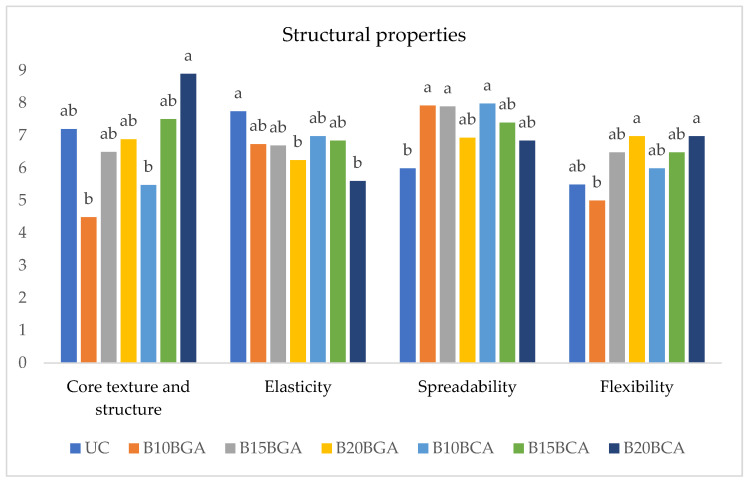
Bread sensory evaluation: structural attributes affected by blueberry variety and fruit addition percentage. Values followed by different letters are significantly different at *p* ≤ 0.05 according to Duncan’s test.

**Figure 8 foods-14-01189-f008:**
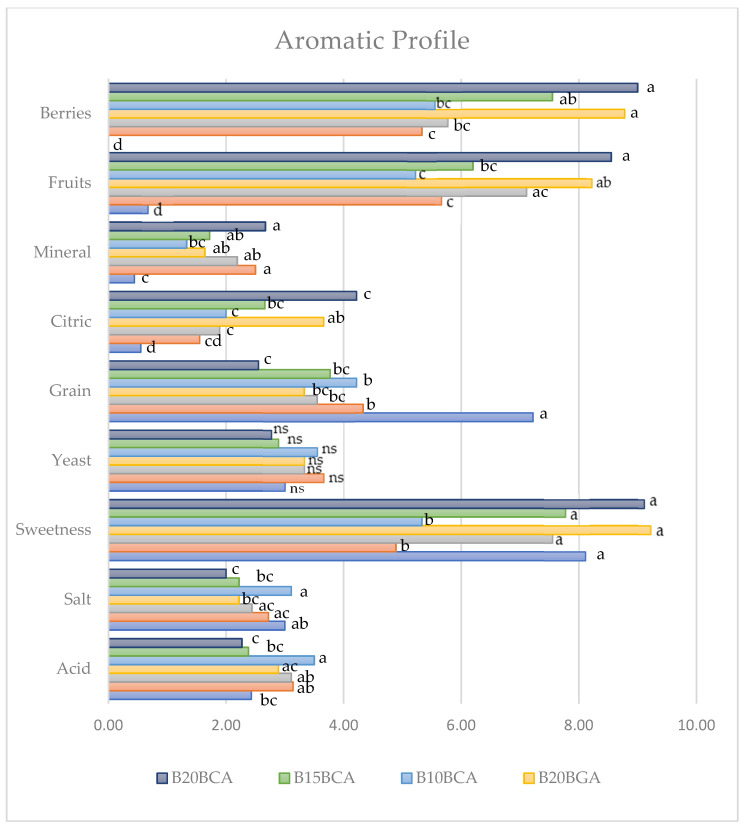
Bread sensory evaluation: aromatic attributes (taste + smell) affected by blueberry variety and fruit addition percentage. Values followed by different letters are significantly different at *p* ≤ 0.05 according to Duncan’s test. ns: not significant.

**Figure 9 foods-14-01189-f009:**
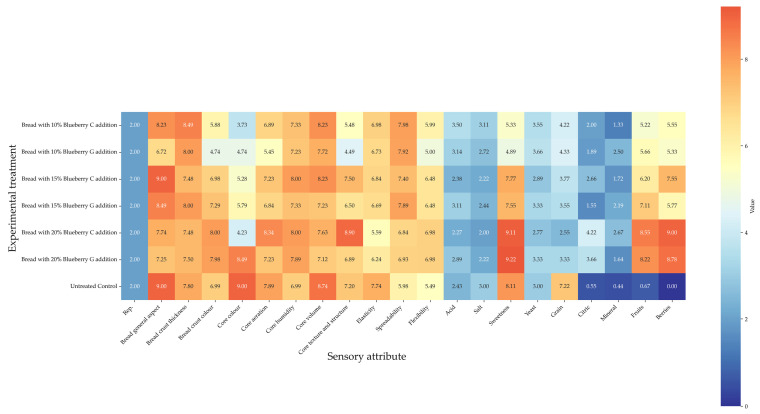
Heatmap related to the effects of the blueberry addition percentages on bread sensory attributes.

**Figure 10 foods-14-01189-f010:**
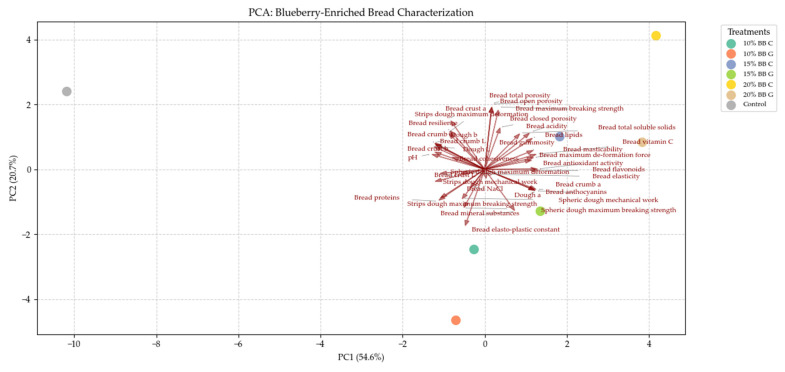
Principal component analysis (PCA).

**Table 1 foods-14-01189-t001:** The recipes for the untreated control and enriched bread.

Ingredients	UC	B10BGA	B15BGA	B20BGA	B10BCA	B15BCA	B20BCA
White flour (%)	60	62	60	59	62	60	59
Water (%)	35	22	21	19.1	22	21	19.1
Yeast (%)	3	3	3	3	3	3	3
Blueberry (%)	-	9.1	10.5	13	9.1	10.5	13
Salt (%)	1	0.9	0.9	0.9	0.9	0.9	0.9
Sugar (%)	1	3	4.5	5	4.5	4.5	5

UC—Untreated Control; B10BGA—bread with 10% blueberry variety G addition; B15BGA—bread with 15% blueberry G addition; B20BGA—bread with 20% blueberry G addition; B10BCA—bread with 10% blueberry variety C addition; B15BCA—bread with 15% blueberry C addition; B20BCA—bread with 20% blueberry C addition; G—Giumalău Mountain harvesting location; C—Ciocănești harvesting location.

**Table 2 foods-14-01189-t002:** Rheological parameters of dough as affected by blueberry variety and fruit addition percentage.

Treatment	Flat Spherical Dough–Bidimensional Elasticity		Flat Strips Dough–Unidimensional Elasticity			
	MBS (N)	MDDS (mm)	MW (mJ)	MBS (N)	MDDS (mm)	MW (mJ)
UC	2.00 ± 0.05 ^ab^	47.32 ± 4.34 ^ab^	49.06 ± 3.21 ^b^	0.73 ± 0.03 ^a^	18.95 ± 1.41 ^a^	8.01 ± 0.38 ^a^
B10BGA	2.40 ± 0.95 ^a^	40.48 ± 8.89 ^c^	54.60 ± 9.20 ^a^	0.65 ± 0.09 ^b^	14.07 ± 1.08 ^bc^	5.25 ± 1.36 ^b^
B15BGA	2.30 ± 0.22 ^a^	43.46 ± 2.54 ^bc^	50.21 ± 7.42 ^ab^	0.63 ± 0.06 ^b^	14.24 ± 1.47 ^b^	5.24 ± 0.87 ^b^
B20BGA	1.72 ± 0.03 ^bc^	48.71 ± 7.68 ^a^	45.87 ± 7.00 ^b^	0.55 ± 0.05 ^c^	14.61 ± 2.86 ^b^	4.63 ± 1.87 ^b^
B10BCA	1.32 ± 0.19 ^cd^	26.98 ± 1.42 ^d^	20.41 ± 2.58 ^c^	0.60 ± 0.01 ^bc^	11.14 ± 1.44 ^c^	4.60 ± 0.17 ^b^
B15BCA	1.20 ± 0.10 ^d^	27.13 ± 5.18 ^d^	19.34 ± 1.22 ^c^	0.47 ± 0.06 ^d^	13.40 ± 0.10 ^b^	3.72 ± 0.90 ^c^
B20BCA	1.17 ± 0.10 ^d^	28.81 ± 1.30 ^d^	16.63 ± 2.70 ^c^	0.45 ± 0.05 ^d^	14.80 ± 1.35 ^b^	2.48 ± 0.56 ^d^

UC—Untreated Control; B10BGA—bread with 10% addition of blueberry fruit variety G; B15BGA—bread with 15% addition of blueberry G; B20BGA—bread with 20% addition of blueberry G; B10BCA—bread with 10% addition of blueberry fruit variety C; B15BCA—bread with 15% addition of blueberry C; B20BCA—bread with 20% addition of blueberry C; G—Giumalău Mountain harvesting location; C—Ciocănești harvesting location; MBS—Maximum breaking strength; MDDS—Maximum deformation of dough sheet; MW—Mechanical work. Within each column, values followed by different letters are significantly different at *p* ≤ 0.05 according to Duncan’s test.

**Table 3 foods-14-01189-t003:** Porosity and rheological parameters of bread as affected by blueberry variety and fruit addition percentage.

Treatment	Total Porosity (%)	Open (%)	Closed (%)	Maximum Breaking Strength (N)	Elasto-Plastic Constant	Maximum Deformation Force (N)
UC	48.7 ± 2.94 ^bc^	47.5 ± 3.28 ^b^	1.19 ± 0.12 ^bc^	12.1 ± 2.2 ^ac^	0.43 ± 0.02 ^a^	11.3 ± 0.7 ^c^
B10BGA	35.7 ± 1.72 ^e^	34.7 ± 2.05 ^d^	1.03 ± 0.14 ^cd^	8.6 ± 3.0 ^c^	0.54 ± 0.06 ^a^	15.3 ± 1.3 ^ab^
B15BGA	41.9 ± 2.14 ^d^	40.6 ± 2.26 ^c^	1.27 ± 0.08 ^ab^	10.9 ± 1.6 ^bc^	0.41 ± 0.03 ^bc^	17.6 ± 1.4 ^a^
B20BGA	44.7 ± 2.66 ^cd^	43.2 ± 2.17 ^c^	1.48 ± 0.18 ^a^	13.3 ± 1.4 ^ab^	0.36 ± 0.06 ^bc^	17.8 ± 2.4 ^a^
B10BCA	42.6 ± 1.64 ^d^	41.8 ± 1.70 ^c^	0.84 ± 0.16 ^d^	10.5 ± 1.9 ^bc^	0.47 ± 0.05 ^ab^	13.5 ± 0.8 ^bc^
B15BCA	52.3 ± 2.83 ^b^	51.3 ± 2.86 ^b^	1.02 ± 0.11 ^cd^	10.6 ± 0.8 ^bc^	0.45 ± 0.07 ^ac^	14.2 ± 1.3 ^bc^
B20BCA	57.3 ± 1.41 ^a^	55.9 ± 1.62 ^a^	1.38 ± 0.08 ^ab^	15.2 ± 1.6 ^a^	0.33 ± 0.02 ^c^	18.5 ± 2.4 ^a^

UC—Untreated Control; B10BGA—bread with 10% addition of blueberry fruit variety G; B15BGA—bread with 15% addition of blueberry G; B20BGA—bread with 20% addition of blueberry G; B10BCA—bread with 10% addition of blueberry fruit variety C; B15BCA—bread with 15% addition of blueberry C; B20BCA—bread with 20% addition of blueberry C; G—Giumalău Mountain harvesting location; C—Ciocănești harvesting location; MBS—Maximum breaking strength; MDDS—Maximum deformation of dough sheet; MW—Mechanical work. Within each column, values followed by different letters are significantly different at *p* ≤ 0.05 according to Duncan’s test.

**Table 4 foods-14-01189-t004:** Rheological parameters of bread as affected by blueberry variety and fruit addition percentage.

Treatment	Firmness (N)	Elasticity	Resilience	Gumminess (N)	Chewiness (N)	Cohesiveness
UC	5.63 ± 0.20 ^a^	0.95 ± 0.03	0.47 ± 0.05 ^a^	5.51 ± 0.46 ^b^	4.42 ± 1.10 ^c^	0.49 ± 0.01
B10BGA	5.45 ± 0.18 ^ab^	0.96 ± 0.02	0.40 ± 0.04 ^bc^	6.72 ± 0.57 ^ab^	6.45 ± 0.59 ^ab^	0.44 ± 0.01
B15BGA	5.30 ± 0.15^ab^	0.93 ± 0.01	0.40 ± 0.09 ^bc^	7.25 ± 0.85 ^ab^	6.97 ± 0.89 ^ab^	0.43 ± 0.02
B20BGA	5.18 ± 0.13 ^b^	0.94 ± 0.01	0.38 ± 0.09 ^bc^	7.46 ± 0.30 ^a^	6.79 ± 0.36 ^ab^	0.41 ± 0.01
B10BCA	5.38 ± 0.16 ^ab^	0.96 ± 0.02	0.37 ± 0.04 ^c^	5.99 ± 0.20 ^ab^	5.15 ± 1.11 ^bc^	0.45 ± 0.02
B15BCA	5.16 ± 0.12 ^b^	0.98 ± 0.01	0.38 ± 0.01 ^bc^	7.18 ± 1.03 ^ab^	7.02 ± 1.15 ^ab^	0.50 ± 0.04
B20BCA	5.09 ± 0.16 ^b^	0.96 ± 0.01	0.43 ± 0.02 ^ab^	7.88 ± 0.98 ^a^	7.52 ± 0.93 ^a^	0.43 ± 0.01
		n.s.				n.s.

UC—Untreated Control; B10BGA—bread with 10% addition of blueberry fruit variety G; B15BGA—bread with 15% addition of blueberry G; B20BGA—bread with 20% addition of blueberry G; B10BCA—bread with 10% addition of blueberry fruit variety C; B15BCA—bread with 15% addition of blueberry C; B20BCA—bread with 20% addition of blueberry C; G—Giumalău Mountain harvesting location; C—Ciocănești harvesting location; MBS—Maximum breaking strength; MDDS—Maximum deformation of dough sheet; MW—Mechanical work. n.s.: not significant; within each column, values followed by different letters are significantly different at *p* ≤ 0.05 according to Duncan’s test.

**Table 5 foods-14-01189-t005:** Quality parameters of bread as affected by blueberry variety and fruitaddition percentage.

Treatment	Acidity (%)	pH	Proteins (%)	Lipids (%)	M.S. (%)	T.S.S. (%)	NaCl (%)
UC	1.47 ± 0.12 ^d^	5.67 ± 0.01 ^a^	13.4 ± 0.6	0.18 ± 0.02 ^d^	1.12 ± 0.12 ^b^	12 ± 1 ^b^	1.17 ± 0.03 ^ab^
B10BGA	1.53 ± 0.12 ^cd^	4.87 ± 0.01 ^b^	13.4 ± 0.4	0.21 ± 0.03 ^cd^	1.37 ± 0.18 ^a^	12 ± 1 ^b^	1.29 ± 0.04 ^a^
B15BGA	1.87 ± 0.12 ^b^	4.84 ± 0.01 ^bc^	13.3 ± 0.5	0.25 ± 0.02 ^bc^	1.23 ± 0.04 ^ab^	13 ± 1 ^b^	1.13 ± 0.12 ^ab^
B20BGA	2.00 ± 0.10 ^ab^	4.76 ± 0.03 ^de^	13.2 ± 0.5	0.30 ± 0.03 ^ab^	1.26 ± 0.02 ^ab^	20 ± 1 ^a^	1.17 ± 0.10 ^ab^
B10BCA	1.67 ± 0.12 ^c^	4.83 ± 0.03 ^bc^	13.3 ± 0.6	0.22 ± 0.02 ^cd^	1.32 ± 0.05 ^ab^	13 ± 1 ^b^	0.91 ± 0.07 ^bc^
B15BCA	2.00 ± 0.10 ^ab^	4.80 ± 0.01 ^cd^	13.3 ± 0.5	0.27 ± 0.03 ^ab^	1.20 ± 0.14 ^ab^	13 ± 1 ^b^	0.91 ± 0.03 ^bc^
B20BCA	2.13 ± 0.12 ^a^	4.72 ± 0.02 ^e^	13.2 ± 0.4	0.31 ± 0.03 ^a^	1.29 ± 0.03 ^ab^	20 ± 1 ^a^	1.06 ± 0.02 ^bc^
			n.s.				

UC—Untreated Control; B10BGA—bread with 10% addition of blueberry fruit variety G; B15BGA—bread with 15% addition of blueberry G; B20BGA—bread with 20% addition of blueberry G; B10BCA—bread with 10% addition of blueberry fruit variety C; B15BCA—bread with 15% addition of blueberry C; B20BCA—bread with 20% addition of blueberry C; G—Giumalău Mountain harvesting location; C—Ciocănești harvesting location; M.S.—Mineral Substances; T.S.S.—Total Soluble Solids. n.s.: not significant; within each column, values followed by different letters are significantly different at *p* ≤ 0.05 according to Duncan’s test.

**Table 6 foods-14-01189-t006:** Bread color parameters as affected by blueberry variety and fruit addition percentage, and production stage.

Treatment	Color Components		
	L	a*	b*
Blueberry var. fruit addition percentage			
UC	71.0 ± 15.7 ^a^	4.6 ± 8.4 ^e^	25.4 ± 3.9 ^a^
B10BGA	12.8 ± 41.5 ^c^	11.8 ± 2.0 ^c^	3.1 ± 11.6 ^d^
B15BGA	11.2 ± 36.5 ^de^	12.4 ± 1.1 ^b^	2.2 ± 10.3 ^e^
B20BGA	10.0 ± 32.9 ^e^	12.0 ± 0.9 ^bc^	0.9 ± 8.0 ^f^
B10BCA	15.3 ± 44.7 ^b^	11.3 ± 0.5 ^d^	4.3 ± 11.2 ^b^
B15BCA	11.8 ± 41.5 ^cd^	12.1 ± 1.2 ^bc^	3.6 ± 10.7 ^cd^
B20BCA	12.3 ± 42.4 ^cd^	13.1 ± 1.7 ^a^	3.9 ± 11.2 ^bc^
Production stage			
Dough	42.9 ± 17.5 ^a^	11.1 ± 5.0 ^b^	−3.0 ± 15.3 ^c^
Bread Crust	42.6 ± 5.0 ^a^	12.9 ± 1.4 ^a^	18.3 ± 5.7 ^a^
Bread Crumb	−23.6 ± 43.5 ^b^	9.1 ± 4.5 ^c^	3.2 ± 12.1 ^b^

UC—Untreated Control; B10BGA—bread with 10% addition of blueberry fruit variety G; B15BGA—bread with 15% addition of blueberry G; B20BGA—bread with 20% addition of blueberry G; B10BCA—bread with 10% addition of blueberry fruit variety C; B15BCA—bread with 15% addition of blueberry C; B20BCA—bread with 20% addition of blueberry C; G—Giumalău Mountain harvesting location; C—Ciocănești harvesting location. Within each column, values followed by different letters are significantly different at *p* ≤ 0.05 according to Duncan’s test.

**Table 7 foods-14-01189-t007:** Antioxidant parameters of bread as affected by blueberry variety and fruit addition percentage.

Treatment	Vitamin C (mg 100 g^−1^)	Total Polyphenols (mg GAE g^−1^)	Flavonoids (mg CE g^−1^)	Total Anthocyanins (mg g^−1^)
UC	0.0 ± 0.0 ^f^	0.63 ± 0.01 ^f^	0.37 ± 0.01 ^e^	0.0 ± 0.0 ^e^
B10BGA	14.1 ± 0.03 ^e^	0.84 ± 0.04 ^c^	0.70 ± 0.01 ^d^	85.3 ± 1.8 ^c^
B15BGA	28.1 ± 0.04 ^c^	1.05 ± 0.04 ^b^	0.83 ± 0.01 ^bc^	100.1 ± 0.5 ^b^
B20BGA	29.7 ± 0.18 ^b^	1.16 ± 0.01 ^a^	0.93 ± 0.01 ^a^	115.3 ± 2.8 ^a^
B10BCA	15.8 ± 0.08 ^d^	0.69 ± 0.02 ^e^	0.71 ± 0.01 ^d^	76.1 ± 3.1 ^d^
B15BCA	29.9 ± 0.06 ^b^	0.76 ± 0.04 ^d^	0.79 ± 0.01 ^c^	81.4 ± 4.3 ^c^
B20BCA	31.6 ± 0.06 ^a^	0.80 ± 0.02 ^cd^	0.86 ± 0.01 ^b^	85.3 ± 2.1 ^c^

UC—Untreated Control; B10BGA—bread with 10% addition of blueberry fruit variety G; B15BGA—bread with 15% addition of blueberry G; B20BGA—bread with 20% addition of blueberry G; B10BCA—bread with 10% addition of blueberry fruit variety C; B15BCA—bread with 15% addition of blueberry C; B20BCA—bread with 20% addition of blueberry C; G—Giumalău Mountain harvesting location; C—Ciocănești harvesting location. Within each column, values followed by different letters are significantly different at *p* ≤ 0.05 according to Duncan’s test.

**Table 8 foods-14-01189-t008:** General aspect, and bread crust and crumb characteristics as affected by blueberry variety and fruit addition percentage.

Treatment	General Aspect	Crust Thickness	Crust Color	Core Color	Core Aeration	Core Humidity	Core Volume
UC	9.00 ± 0.00 ^a^	7.80 ± 1.04 ^ab^	6.99 ± 0.96 ^b^	9.00 ± 0.00 ^a^	7.89 ± 0.97 ^a^	6.99 ± 1.06	8.75 ± 0.44
B10BGA	6.72 ± 0.94 ^d^	8.00 ± 1.06 ^ab^	4.74 ± 0.92 ^d^	4.74 ± 0.42 ^b^	5.45 ± 0.82 ^b^	7.23 ± 1.14	7.73 ± 0.44
B15BGA	8.49 ± 0.88 ^ab^	8.00 ± 0.96 ^ab^	7.29 ± 0.59 ^ab^	5.79 ± 0.76 ^b^	6.84 ± 1.11 ^ab^	7.34 ± 1.29	7.23 ± 1.14
B20BGA	7.25 ± 0.83 ^cd^	7.50 ± 1.13 ^ab^	7.98 ± 0.88 ^a^	8.49 ± 0.88 ^a^	7.23 ± 1.01 ^ab^	7.89 ± 1.32	7.13 ± 1.01
B10BCA	8.24 ± 0.77 ^ab^	8.50 ± 0.88 ^a^	5.88 ± 1.05 ^c^	3.74 ± 1.09 ^b^	6.89 ± 0.79 ^ab^	7.34 ± 1.29	8.24 ± 0.77
B15BCA	9.00 ± 0.00 ^a^	7.49 ± 1.06 ^b^	6.98 ± 0.86 ^b^	5.28 ± 1.07 ^b^	7.23 ± 0.42 ^ab^	8.00 ± 1.37	8.24 ± 0.77
B20BCA	7.74 ± 0.83 ^bc^	7.49 ± 1.06 ^b^	8.00 ± 1.06 ^a^	4.23 ± 1.04 ^b^	8.34 ± 0.79 ^a^	8.00 ± 1.37	7.64 ± 1.09
						n.s.	n.s.

UC—Untreated Control; B10BGA—bread with 10% addition of blueberry fruit variety G; B15BGA—bread with 15% addition of blueberry G; B20BGA—bread with 20% addition of blueberry G; B10BCA—bread with 10% addition of blueberry fruit variety C; B15BCA—bread with 15% addition of blueberry C; B20BCA—bread with 20% addition of blueberry C; G—Giumalău Mountain harvesting location; C—Ciocănești harvesting location. n.s.: not significant; within each column, values followed by different letters are significantly different at *p* ≤ 0.05 according to Duncan’s test.

## Data Availability

The original contributions presented in this study are included in the article. Further inquiries can be directed to the corresponding authors.

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
