# Peer review of "Effect of Wheat Flour Integration with Blueberry Fruits on Rheological, Quality, Antioxidant, and Sensory Attributes of ‘French’ Bread"

_foods, 2025, doi:10.3390/foods14071189_

Round 1
Reviewer 1 Report
Comments and Suggestions for Authors
This paper addresses the fortification of wheat bread with blueberries, aligning with current trends in functional foods and health-conscious consumer preferences. Also, it contributes to sustainable food development by incorporating natural antioxidants into a widely consumed product. The study is well-structured and scientifically rigorous, using multiple treatments and the inclusion of a control group for meaningful comparisons. It also examines multiple quality parameters, including rheological, textural and sensorial properties, as well as antioxidant capacity determination. The obtained results reinforce the potential health benefits of consuming fortified bread. Here are some suggestions:
Authors should consider the cost implications of blueberry fortification, which is essential for assessing the commercial viability of such a product. Also, there is no mention of how blueberry addition affects the shelf-life of the bread in terms of microbial stability, texture changes over time, or oxidative deterioration. Since blueberries contain moisture and bioactive compounds, the storage stability is an important parameter to evaluate.
While sensory evaluation was conducted, the panel consisted of only 20 participants. For that, a larger and more diverse consumer panel would strengthen the validity of sensory acceptance findings.
The extensive use of tables a numerical data makes the study difficulte to navigate for readers, I could recommend summarize key findings with concise charts, which could enhance readability.
The study emphasizes in antioxidant benefits, however, other functional properties such a dietary fiber content, glycemix index changes, or potential prebiotic effects of blueberry-fortified bread are not explored.
While the study emphasizes antioxidant benefits, other functional properties such as dietary fiber content, glycemic index changes, or potential prebiotic effects of blueberry-fortified bread are not explored.
Specific comments:
Throughout the manuscript, replace “hours” with “h”; “minutes” by “min”, “ml” by “mL”, including fig. 4.
Line 20: Replace “10 %; 15 % and 20 %” with “10, 15 and 20 %
Line 23: Replace “10 % and 15 %” with “10 and 15 %
Line 32: Replace “10 % to 20 15 %” with “10 to 20 %
Line 135, 151: Replace “10 %; 15 % and 20 %” with “10, 15 and 20 %
Line 381, 384: Replace “10 % and 15 %” with “10 and 15 %
Line 392, 525, 608, 643: Replace “10 % and 20 %” with “10 and 20 %
Line 423: Replace “5.9 % and 19.9 %” with “5.9 and 19.9 %
Line 424: Replace “7.8 % and 9.0 %” with “7.8 and 9.0 %
Line 542-543: Replace “4 % to 10 %, “5 % to 20%, and “10 % to 30 %” with “4 to 10 %, “5 to 20%, and “10 to 30 %” respectively.
Line 559: Replace“113 %, 63 % and 68 %” with “113, 63 and 68 %”.
Line 623: Put the scientific name “P. ostreatus” in italics.
Reference 31 is incomplete.
Make sure that all scientific names appear in italics.
Author Response
Reviewer Comments
Reviewer 1
Quality of English Language
(x) The English is fine and does not require any improvement.
Yes Can be improved Must be improved Not applicable
Does the introduction provide sufficient background and include all relevant references?
(x) ( ) ( ) ( )
Is the research design appropriate?
(x) ( ) ( ) ( )
Are the methods adequately described?
(x) ( ) ( ) ( )
Are the results clearly presented?
(x) ( ) ( ) ( )
Are the conclusions supported by the results?
(x) ( ) ( ) ( )
Comments and Suggestions for Authors
This paper addresses the fortification of wheat bread with blueberries, aligning with current trends in functional foods and health-conscious consumer preferences. Also, it contributes to sustainable food development by incorporating natural antioxidants into a widely consumed product. The study is well-structured and scientifically rigorous, using multiple treatments and the inclusion of a control group for meaningful comparisons. It also examines multiple quality parameters, including rheological, textural and sensorial properties, as well as antioxidant capacity determination. The obtained results reinforce the potential health benefits of consuming fortified bread.
Here are some suggestions:
Authors should consider the cost implications of blueberry fortification, which is essential for assessing the commercial viability of such a product. Also, there is no mention of how blueberry addition affects the shelf-life of the bread in terms of microbial stability, texture changes over time, or oxidative deterioration. Since blueberries contain moisture and bioactive compounds, the storage stability is an important parameter to evaluate.
While sensory evaluation was conducted, the panel consisted of only 20 participants. For that, a larger and more diverse consumer panel would strengthen the validity of sensory acceptance findings.
The extensive use of tables a numerical data makes the study difficulte to navigate for readers, I could recommend summarize key findings with concise charts, which could enhance readability.
While the study emphasizes antioxidant benefits, other functional properties such as dietary fiber content, glycemic index changes, or potential prebiotic effects of blueberry-fortified bread are not explored.
Answer: Dear Reviewer, we have added the cost implications related to blueberry addition to bread in the Conclusions section.
Answer: We have mentioned in each Materials and Methods subsection that we repeated all the determinations a week after producing the blueberry added bread, but no significant differences arose in comparison with the initial values and, therefore, we did not show the data.
Answer: We chose to involve 20 panellists to evaluate the sensory attributes of blueberry added bread, in comparison with the control, because all of them were appropriately instructed and trained to perform this activity and, in this respect, a higher number of participants would have made our teaching work difficult to be managed.
Answer: We have added a concise figure regarding the results of PCA.
Answer: Dear Reviewer, we have not measured in this study the dietary fiber, glycemic index and potential prebiotic effect of the blueberry added bread, but we are confident to perform the mentioned determinations in the near future.
Specific comments:
Throughout the manuscript, replace “hours” with “h”; “minutes” by “min”, “ml” by “mL”, including fig. 4.
Answer: Addressed.
Line 20: Replace “10 %; 15 % and 20 %” with “10, 15 and 20 %
Answer: Addressed.
Line 23: Replace “10 % and 15 %” with “10 and 15 %
Answer: Addressed.
Line 32: Replace “10 % to 20 15 %” with “10 to 20 %
Answer: Addressed.
Line 135, 151: Replace “10 %; 15 % and 20 %” with “10, 15 and 20 %
Answer: Addressed.
Line 381, 384: Replace “10 % and 15 %” with “10 and 15 %
Answer: Addressed.
Line 392, 525, 608, 643: Replace “10 % and 20 %” with “10 and 20 %
Answer: Addressed.
Line 423: Replace “5.9 % and 19.9 %” with “5.9 and 19.9 %
Answer: Addressed.
Line 424: Replace “7.8 % and 9.0 %” with “7.8 and 9.0 %
Answer: Addressed.
Line 542-543: Replace “4 % to 10 %, “5 % to 20%, and “10 % to 30 %” with “4 to 10 %, “5 to 20%, and “10 to 30 %” respectively.
Answer: Addressed.
Line 559: Replace“113 %, 63 % and 68 %” with “113, 63 and 68 %”.
Answer: Addressed.
Line 623: Put the scientific name “P. ostreatus” in italics.
Answer: Addressed.
Reference 31 is incomplete.
Answer: Addressed.
Make sure that all scientific names appear in italics.
Answer: Addressed.
Reviewer 2 Report
Comments and Suggestions for Authors
GENERAL COMMENTS
Dear authors, congratulations on your manuscript,
Enriching bread with blueberry isn't exactly innovative, but it is an interesting variation on baking. However, the idea of incorporating spontaneous blueberries into a functional approach is something more innovative.
I have some major questions.
Kind Regards,
Reviewer,
SPECIFIC COMMENTS
Line 93 – (Vaccinum myrtillus L.)
Line 161 - Indicate the characteristics of the container in which the blueberries were frozen. Repeat this procedure throughout the manuscript.
Line 167 – and sugar (1%)
Table 1 – Ingredients (%) eliminate % white flour... salt, and sugar.
Table 1 - delete commas (,) and add points (.)
Table 1 - the authors mention 10%, 15% and 20%, but in the table the blueberry percentages are 9.1, 10.5 and 13% respectively?
Line 195 - the authors write Blueberry in practically every paper, but here they write blueberries?
Author Response
Reviewer 2
Effect of Wheat Flour Fortification with Blueberry Fruits on Bread Rheological, Quality, Antioxidant and Sensorial Characteristics
What is the main question addressed by the research?
This research assesses the influence of wheat flour fortification with blueberry fruits on rheological, quality, antioxidant, and sensorial characteristics of bread and investigates how different concentrations (10%, 15%, and 20%) of blueberries from two varieties, which were harvested from mountains in Romania, are affecting bread properties regarding its texture, porosity, color, antioxidant content, and consumer acceptability. Results emphasize the possibility to use blueberry as a functional ingredient for improving the nutritional characteristics of bread while maintaining desirable sensory properties.
Originality/Novelty
The innovation in this study was that it fortified wheat flour with blueberry fruits to improve bread nutritional and functional properties. While the originality due to Use of Wild Blueberries from the Rarău and Ciocănești mountains of Romania, which may have different bioactive compositions comparing to varieties that are commercially cultivated. Also, most of the previous studies have investigated one or two parameters such as antioxidant content or texture but this study covered more parameters like rheological properties of dough, textural and structural changes in bread, antioxidant content, color, and sensory acceptability. Adding various blueberry concentrations provides new knowledge on how to optimize fruit incorporation without compromising major baking properties related to texture, volume, and porosity.
Significance of Content
This work fills the gaps in knowledge by investigating the rheological, nutritional, and sensory properties of bread enriched with wild blueberries, which has not been widely investigated. The findings can be used as a starting point for future studies in functional bakery product development and inspire new applications of fruit-based fortification in food technology.
Did you detect plagiarism?
Yes, 35%
Did you detect inappropriate self-citations by authors?
Yes
in Line 71: self-citation [11], wrong citation
Answer: Addressed.
in Line 90: self-citation [19], the study used fruit not byproduct, wrong citation
Answer: Addressed.
in Line 97: self-citation [21], no need citation
Answer: Addressed.
in Line 125: self-citation [26], it is general information and wrong citation
Answer: Addressed.
in Line 182: self-citation [27], the information given is mentioned in the introduction and has no meaning in
the Materials and Methods section.
Answer: Addressed.
in Line 278: self-citation [30], actually the needed reference is (29) already used.
Answer: Addressed.
Does the introduction provide sufficient background and include all relevant references?
- The introduction discusses general fruit fortification without prior research references specifically on blueberries in bread or bakery products. This could be further strengthened by adding references to previous studies that investigated the effect of blueberry on dough properties, texture, or sensory attributes.
Answer: We have addressed the above recommendations.
- Line 135: There is no explanation in the introduction for why these concentrations were selected.
Answer: The range of blueberry addition percentages was chosen based both on literature reports regarding either the same fruit or similar ones and on the expected technical-economic balance related to the product to manufacture.
Is the research design appropriate?
- The research design is generally appropriate and well-structured for the study's objectives, providing reliable and scientifically valid results. The study measures nutritional and sensory properties, but it does not test how these properties change over time.
Answer: We have mentioned in each Materials and Methods subsection that we repeated all the determinations a week after producing the blueberry added bread, but no significant differences arose in comparison with the initial values and, therefore, we did not show the data.
Are the methods adequately described?
The study provides good descriptions of its methods; however, it needs more detailed information in certain areas as following:
- No deeper examination has been given for gluten network formation, as well as on starch gelatinization.
Answer: Addressed.
- The sensory analysis section talks of 20 panelists but does not outline whether they were trained or untrained.
Answer: The panellists were trained for about a month, three days a week, by explaining them the criteria to evaluate each of the bread attributes examined, benefitting from the instructions given by some professors included in the author list and some experts in this field.
Are the results clearly presented?
The results are scientifically sound and thoroughly documented using tables and statistical analysis. However, some areas could be improved.
- Additionally, incorporating graphical representations or heatmap for the response surface model (RSM) would enhance their clarity and accessibility. For example, A bar graph comparing DPPH antioxidant activity across treatments would visually highlight differences. Also, a color-coded heatmap could show changes in sensory ratings based on blueberry concentration.
Answer: Both a figure showing the DPPH comparisons and a heatmap regarding the sensory attributes have been added.
The results of sensory evaluation are shortly discussed; however, more details on specific consumer preferences would be relevant. For example, Were there any negative sensory effects at high blueberry concentrations? Which attributes were most affected by the fruit integration: taste, color, or texture?
Answer: The blueberry addition had a worse effect, compared to the untreated control, on cereal taste and elasticity perception of bread. On the contrary, increasing integration of V. myrtillus fruits from 10 to 20% enhanced the spreadability, tart, citric, mineral, fruit and berry taste, inner colour, texture, structure and aeration, flexibility, sweetness, and general acceptability.
Are the conclusions supported by the results?
The conclusions generally comply with the findings, and they provide a comprehensive summary of the key findings regarding enhancing bread with blueberries.
However, certain adjustments would enhance clarity as well as overall effectiveness.
- The higher blueberry content affects dough texture and structure—that needs to be clearly mentioned.
Answer: We have mentioned the above outcome, as recommended.
- Some sensory attributes reduced at higher blueberry content—that needs to be mentioned as a limitation.
Answer: We have added the above suggestion.
- No indication of future research, for example: long-term stability of antioxidants when bread is stored. Using different fruit-processing methods (freeze-dried vs. fresh pulp) for fortification. Investigation of consumer acceptability under actual market conditions.
References
- References: 9, 16, 24, 39, 46, 49, 58, 68 the references are too old, please use a recent one.
Answer: Addressed.
- Some standardized methods (e.g., bread preparation, Extraction of Biologically active compound, and determination of bread porosity, rheological properties and color) are described but without references.
Answer: Addressed.
Reviewer 3 Report
Comments and Suggestions for Authors
Please find all comments in attached file

Author Response
Reviewer Comments
Reviewer 3
Yes Can be improved Must be improved Not applicable
Does the introduction provide sufficient background and include all relevant references?
( ) (x) ( ) ( )
Is the research design appropriate?
(x) ( ) ( ) ( )
Are the methods adequately described?
( ) (x) ( ) ( )
Are the results clearly presented?
( ) (x) ( ) ( )
Are the conclusions supported by the results?
( ) ( ) ( ) ( )
Comments and Suggestions for Authors
After reading the manuscript "Effect of Wheat Flour Fortification with Blueberry Fruits on Bread Rheological, Quality, Antioxidant and Sensorial Charateristics", I realized that the manuscript showed in some parts the scientific rigour wanted, but in other parts I have missed it.
The authors have presented critical evaluation only in some paragraphs.
The references are not exactly current, besides material and methods need adjustments.
Thats why I have written some suggestions below in an attempt to improve the paper.
Please, adjust the title on the system : "Insight into the Development of Bakery Products Based on Wheat Flour Fortified with Blueberry Fruits"
L.3- Prefer sensory, instead of "sensorial" - check the whole paper, please.
Answer: Dear Reviewer, we have addressed your recommendation.
L.3- Attributes ? Descriptors ? Check methodology you follow.
Answer: We have replaced ‘Characteristics’ with ‘Attributes’.
L.47- "snacks" for all - Is it correct ? I do not agree.
Answer: We have replaced ‘snacks’ with ‘items’.
L.48- This paragraph is too long.
Answer: We have reduced the paragraph length.
L.133- "two different sources (Giumalău peak, Rarău mountains, and Câmpulung-Suceava) " 2 or 3 ? CiocăneÈ™ti mountains (Suceava) Very confused this paragraph.
Answer: We have made this paragraph clearer.
L.171- please, include more details about the ingredients.
Eg. warm water ? chemical yeast ? demerara sugar ?
Answer: Addressed.
wouldn't the amount of water be a bias?
Answer: We have added the above suggestion.
In the section 2,2 on the line 167 and table, 1 it was specified the amount of water for control sample and of water and blueberry juice for the other samples. Also it these sections could be observed the proportion of all the ingredients used in the production process
L.204- I couldn't find out whether you used the direct or indirect method for the dough.
Answer: We have addressed the above suggestion.
L.234- how many batches?
Answer: We have addressed the above suggestion.
How were the batches repeated? Same batch or different batches?
Six types of bread (B) enriched with blueberries and one untreated control with three batches in the same ambient conditions were manufactured according to the recipes shown in Table 1 which, respect to the technological flow, includes the following stages: Scaling – Mixing – Kneading and folding – Fermentation – Shaping – Rising – Baking – Cooling (Figure 3 and 4).
I think it's important to highlight the type of bread chosen for evaluation, there are many types of bread in the literature.
Answer: We have addressed the above suggestion.
L.236- Please, improve Figure 3
Answer: We have addressed the above suggestion.
L.250- Analysis or analyses, please check English
Answer: Addressed.
L.282- How many replicates, please ?
Answer: We have added the number of replicates.
L.297- TMA - It seems to me that some acronyms were not used throughout the work, such as TMA and TPC.
L.310 - TPC
Answer: We have deleted the mentioned acronyms everywhere in the text.
L.340- "cohesiveness, elasticity, gumminess, chewiness and resilience" X table 4 - Elasticity Resilience, Gummosity (N) Masticability , (N) Cohesiveness - please corret. I missed the other Units as well.
Answer: We have uniformed the sequence of labels and added the missing Units.
347-What about firmness result ?
Answer: Addressed.
L.353- For a sensory test a lot of relevant information was not included, I suggest reading papers and improving yours.
Answer: Addressed.
Has the project been submitted to an evaluation by a university ethics committee? Did it follow the Helsinki declaration? Please, enter the approval protocol number. Which sensory test was performed? Since the assessors were trained, how many sessions were conducted ? What was the profile of the assessors ? Were the analyses performed in sensory booths ? How many men/women ? Are the assessors usually consumers of this product ?
Answer: Addressed.
L.359- " a 2 point scale referring to some characteristics" - I am sorry, but I do not understand. 2 point ? Authors, please.
"characteristics" - attribute.
Answer: We have replaced ‘characteristics’ with ‘attributes’.
L.360 " appearance, texture, elasticity, texture and structure of the core, spreadability (by friction), flexibility (by bending)", but on line L.590 " General aspect Crust thickness Crust colour Core colour Core aeration Core humidity Core volume, Please correct. What were the attibutes evaluated ? Very confused.
Answer: Addressed.
What about taste and odor, authors ?
Answer: We have addressed the above suggestion.
L.361- "a 15 point hedonic scale for the 361 main aromas" - authors/ year
Answer: Addressed.
L.363- "For all parameters under evaluation, 0 indicates extreme dislike and the maxim value (2, 363 15 or 9), represents atmost liking" - authors/ year
Answer: We have addressed the above suggestion.
L.458- Page 12 is blank. I missed the paragraphs in the whole paper.The letters used in tables for comparisons of means are more delicate in superscript.
Answer: We have addressed the above comment.
L.462- I am sorry but i did not find in Material and Methods"MBS (N) MDDS (mm) MW (mJ) MBS (N) MDDS (mm) MW (mJ)"
Answer: We have addressed the above suggestion.
L.469- I am sorry but i did not find as well in Material and Methods - "Open (%) Closed (%) Maximum breaking strength (N) Elasto-plastic con stant Maximum de formation force "
Answer: We have addressed the above suggestion.
L.512 - I missed the statistical analysis in table 6 for a* and b* color parameters
Answer: We have added the letters to the mentioned colour parameters.
L.520- I think i do not follow the relevance of Fig. 5. Check bread crumb " l l "
Answer: We have reported the Fig. 5 because the significant interaction between blueberry variety fruit addition percentage and production stage on bread colour components arose. We have checked the mentioned letters.
L.528- Authors, is it clear to you about Vitamin C in the Material and Methods?
Answer: We have added the method related to vitamin C determination.
L.544- Poiana et al. [5] - It is not 5, but 35
Answer: Corrected.
L.601- Did you evaluate taste ? Can tou state the paragrpah below ?
"the lowest spreadability, tart, citric, mineral, fruit and berry taste; not significantly difference from the blueberry addition treatments generally regarding inner texture and structure, flexibility and salt taste; lower acid taste than the 10% variety C blueberry addition. As for the comparison between the blueberry addition treatments, in most cases (inner texture and structure, flexibility, sweetness, citric, fruit and berry taste, inner colour and aeration, general acceptability and degree of general acceptability) a general increasing trend with the integration rising from 10% to 20% was recorded. Opposite trend was shown by elasticity, acid, salty and cereal taste. The trends of tart and mineral taste were controversial."
Answer: Addressed.
L.617- Check Figure 7 - Eg. Sweetnes and Salt- Check English
Answer: Corrected.
L.641- It seems to me that you have some very interesting findings that were not covered in the conclusion, only in a very general approach. For example, the findings in table 7.
Answer: We have added the significant effect of blueberry integration into bread on the antioxidant compounds and activity, as recommended.
Comments on the Quality of English Language
The English could be improved to more clearly express the research.
Answer: Addressed.
Reviewer 4 Report
Comments and Suggestions for Authors
After reading the manuscript "Effect of Wheat Flour Fortification with Blueberry Fruits on Bread Rheological, Quality, Antioxidant and Sensorial Charateristics", I realized that the manuscript showed in some parts the scientific rigour wanted, but in other parts I have missed it.
The authors have presented critical evaluation only in some paragraphs.
The references are not exactly current, besides material and methods need adjustments.
Thats why I have written some suggestions below in an attempt to improve the paper.
Please, adjust the title on the system : "Insight into the Development of Bakery Products Based on Wheat Flour Fortified with Blueberry Fruits"
L.3- Prefer sensory, instead of "sensorial" - check the whole paper, please.
L.3- Attributes ? Descriptors ? Check methodology you follow.
L.47- "snacks" for all - Is it correct ? I do not agree.
L.48- This paragraph is too long.
L.133- "two different sources (Giumalău peak, Rarău mountains, and Câmpulung-Suceava) " 2 or 3 ? CiocăneÈ™ti mountains (Suceava) Very confused this paragraph.
L.171- please, include more details about the ingredients.
Eg. warm water ? chemical yeast ? demerara sugar ?
wouldn't the amount of water be a bias?
L.204- I couldn't find out whether you used the direct or indirect method for the dough.
L.234- how many batches?
How were the batches repeated? Same batch or different batches?
I think it's important to highlight the type of bread chosen for evaluation, there are many types of bread in the literature.
L.236- Please, improve Figure 3
L.250- Analysis or analyses, please check English
L.282- How many replicates, please ?
L.297- TMA - It seems to me that some acronyms were not used throughout the work, such as TMA and TPC.
L.310 - TPC
L.340- "cohesiveness, elasticity, gumminess, chewiness and resilience" X table 4 - Elasticity Resilience, Gummosity (N) Masticability , (N) Cohesiveness - please corret. I missed the other Units as well.
347-What about firmness result ?
L.353- For a sensory test a lot of relevant information was not included, I suggest reading papers and improving yours.
Has the project been submitted to an evaluation by a university ethics committee? Did it follow the Helsinki declaration? Please, enter the approval protocol number. Which sensory test was performed? Since the assessors were trained, how many sessions were conducted ? What was the profile of the assessors ? Were the analyses performed in sensory booths ? How many men/women ? Are the assessors usually consumers of this product ?
L.359- " a 2 point scale referring to some characteristics" - I am sorry, but I do not understand. 2 point ? Authors, please.
"characteristics" - attribute.
L.360 " appearance, texture, elasticity, texture and structure of the core, spreadability (by friction), flexibility (by bending)", but on line L.590 " General aspect Crust thickness Crust colour Core colour Core aeration Core humidity Core volume, Please correct. What were the attibutes evaluated ? Very confused.
What about taste and odor, authors ?
L.361- "a 15 point hedonic scale for the 361 main aromas" - authors/ year
L.363- "For all parameters under evaluation, 0 indicates extreme dislike and the maxim value (2, 363 15 or 9), represents atmost liking" - authors/ year
L.458- Page 12 is blank. I missed the paragraphs in the whole paper.The letters used in tables for comparisons of means are more delicate in superscript.
L.462- I am sorry but i did not find in Material and Methods"MBS (N) MDDS (mm) MW (mJ) MBS (N) MDDS (mm) MW (mJ)"
L.469- I am sorry but i did not find as well in Material and Methods - "Open (%) Closed (%) Maximum breaking strength (N) Elasto-plastic con stant Maximum de formation force "
L.512 - I missed the statistical analysis in table 6 for a* and b* color parameters
L.520- I think i do not follow the relevance of Fig. 5. Check bread crumb " l l "
L.528- Authors, is it clear to you about Vitamin C in the Material and Methods?
L.544- Poiana et al. [5] - It is not 5, but 35
L.601- Did you evaluate taste ? Can tou state the paragrpah below ?
"the lowest spreadability, tart, citric, mineral, fruit and berry taste; not significantly difference from the blueberry addition treatments generally regarding inner texture and structure, flexibility and salt taste; lower acid taste than the 10% variety C blueberry addition. As for the comparison between the blueberry addition treatments, in most cases (inner texture and structure, flexibility, sweetness, citric, fruit and berry taste, inner colour and aeration, general acceptability and degree of general acceptability) a general increasing trend with the integration rising from 10% to 20% was recorded. Opposite trend was shown by elasticity, acid, salty and cereal taste. The trends of tart and mineral taste were controversial."
L.617- Check Figure 7 - Eg. Sweetnes and Salt- Check English
L.641- It seems to me that you have some very interesting findings that were not covered in the conclusion, only in a very general approach. For example, the findings in table 7.
Comments on the Quality of English Language
The English could be improved to more clearly express the research.
Author Response
Reviewer Comments
Reviewer 4
Yes Can be improved Must be improved Not applicable
Does the introduction provide sufficient background and include all relevant references?
( ) (x) ( ) ( )
Is the research design appropriate?
(x) ( ) ( ) ( )
Are the methods adequately described?
( ) (x) ( ) ( )
Are the results clearly presented?
( ) (x) ( ) ( )
Are the conclusions supported by the results?
( ) ( ) ( ) ( )
Comments and Suggestions for Authors
After reading the manuscript "Effect of Wheat Flour Fortification with Blueberry Fruits on Bread Rheological, Quality, Antioxidant and Sensorial Charateristics", I realized that the manuscript showed in some parts the scientific rigour wanted, but in other parts I have missed it.
The authors have presented critical evaluation only in some paragraphs.
The references are not exactly current, besides material and methods need adjustments.
Thats why I have written some suggestions below in an attempt to improve the paper.
Please, adjust the title on the system : "Insight into the Development of Bakery Products Based on Wheat Flour Fortified with Blueberry Fruits"
L.3- Prefer sensory, instead of "sensorial" - check the whole paper, please.
Answer: Dear Reviewer, we have addressed your recommendation.
L.3- Attributes ? Descriptors ? Check methodology you follow.
Answer: We have replaced ‘Characteristics’ with ‘Attributes’.
L.47- "snacks" for all - Is it correct ? I do not agree.
Answer: We have replaced ‘snacks’ with ‘items’.
L.48- This paragraph is too long.
Answer: We have reduced the paragraph length.
L.133- "two different sources (Giumalău peak, Rarău mountains, and Câmpulung-Suceava) " 2 or 3 ? CiocăneÈ™ti mountains (Suceava) Very confused this paragraph.
Answer: We have made this paragraph clearer.
L.171- please, include more details about the ingredients.
Eg. warm water ? chemical yeast ? demerara sugar ?
Answer: Addressed.
wouldn't the amount of water be a bias?
Answer: We have added the above suggestion.
In the section 2,2 on the line 167 and table, 1 it was specified the amount of water for control sample and of water and blueberry juice for the other samples. Also it these sections could be observed the proportion of all the ingredients used in the production process
L.204- I couldn't find out whether you used the direct or indirect method for the dough.
Answer: We have addressed the above suggestion.
L.234- how many batches?
Answer: We have addressed the above suggestion.
How were the batches repeated? Same batch or different batches?
Six types of bread (B) enriched with blueberries and one untreated control with three batches in the same ambient conditions were manufactured according to the recipes shown in Table 1 which, respect to the technological flow, includes the following stages: Scaling – Mixing – Kneading and folding – Fermentation – Shaping – Rising – Baking – Cooling (Figure 3 and 4).
I think it's important to highlight the type of bread chosen for evaluation, there are many types of bread in the literature.
Answer: We have addressed the above suggestion.
L.236- Please, improve Figure 3
Answer: We have addressed the above suggestion.
L.250- Analysis or analyses, please check English
Answer: Addressed.
L.282- How many replicates, please ?
Answer: We have added the number of replicates.
L.297- TMA - It seems to me that some acronyms were not used throughout the work, such as TMA and TPC.
L.310 - TPC
Answer: We have deleted the mentioned acronyms everywhere in the text.
L.340- "cohesiveness, elasticity, gumminess, chewiness and resilience" X table 4 - Elasticity Resilience, Gummosity (N) Masticability , (N) Cohesiveness - please corret. I missed the other Units as well.
Answer: We have uniformed the sequence of labels and added the missing Units.
347-What about firmness result ?
Answer: Addressed.
L.353- For a sensory test a lot of relevant information was not included, I suggest reading papers and improving yours.
Answer: Addressed.
Has the project been submitted to an evaluation by a university ethics committee? Did it follow the Helsinki declaration? Please, enter the approval protocol number. Which sensory test was performed? Since the assessors were trained, how many sessions were conducted ? What was the profile of the assessors ? Were the analyses performed in sensory booths ? How many men/women ? Are the assessors usually consumers of this product ?
Answer: Addressed.
L.359- " a 2 point scale referring to some characteristics" - I am sorry, but I do not understand. 2 point ? Authors, please.
"characteristics" - attribute.
Answer: We have replaced ‘characteristics’ with ‘attributes’.
L.360 " appearance, texture, elasticity, texture and structure of the core, spreadability (by friction), flexibility (by bending)", but on line L.590 " General aspect Crust thickness Crust colour Core colour Core aeration Core humidity Core volume, Please correct. What were the attibutes evaluated ? Very confused.
Answer: Addressed.
What about taste and odor, authors ?
Answer: We have addressed the above suggestion.
L.361- "a 15 point hedonic scale for the 361 main aromas" - authors/ year
Answer: Addressed.
L.363- "For all parameters under evaluation, 0 indicates extreme dislike and the maxim value (2, 363 15 or 9), represents atmost liking" - authors/ year
Answer: We have addressed the above suggestion.
L.458- Page 12 is blank. I missed the paragraphs in the whole paper.The letters used in tables for comparisons of means are more delicate in superscript.
Answer: We have addressed the above comment.
L.462- I am sorry but i did not find in Material and Methods"MBS (N) MDDS (mm) MW (mJ) MBS (N) MDDS (mm) MW (mJ)"
Answer: We have addressed the above suggestion.
L.469- I am sorry but i did not find as well in Material and Methods - "Open (%) Closed (%) Maximum breaking strength (N) Elasto-plastic con stant Maximum de formation force "
Answer: We have addressed the above suggestion.
L.512 - I missed the statistical analysis in table 6 for a* and b* color parameters
Answer: We have added the letters to the mentioned colour parameters.
L.520- I think i do not follow the relevance of Fig. 5. Check bread crumb " l l "
Answer: We have reported the Fig. 5 because the significant interaction between blueberry variety fruit addition percentage and production stage on bread colour components arose. We have checked the mentioned letters.
L.528- Authors, is it clear to you about Vitamin C in the Material and Methods?
Answer: We have added the method related to vitamin C determination.
L.544- Poiana et al. [5] - It is not 5, but 35
Answer: Corrected.
L.601- Did you evaluate taste ? Can tou state the paragrpah below ?
"the lowest spreadability, tart, citric, mineral, fruit and berry taste; not significantly difference from the blueberry addition treatments generally regarding inner texture and structure, flexibility and salt taste; lower acid taste than the 10% variety C blueberry addition. As for the comparison between the blueberry addition treatments, in most cases (inner texture and structure, flexibility, sweetness, citric, fruit and berry taste, inner colour and aeration, general acceptability and degree of general acceptability) a general increasing trend with the integration rising from 10% to 20% was recorded. Opposite trend was shown by elasticity, acid, salty and cereal taste. The trends of tart and mineral taste were controversial."
Answer: Addressed.
L.617- Check Figure 7 - Eg. Sweetnes and Salt- Check English
Answer: Corrected.
L.641- It seems to me that you have some very interesting findings that were not covered in the conclusion, only in a very general approach. For example, the findings in table 7.
Answer: We have added the significant effect of blueberry integration into bread on the antioxidant compounds and activity, as recommended.
Comments on the Quality of English Language
The English could be improved to more clearly express the research.
Answer: Addressed.

Round 2
Reviewer 3 Report
Comments and Suggestions for Authors
please see the attached file

Comments on the Quality of English Language
The language of the research still needs improvement, and some sentences contain grammatical errors and several logical order of discussion.
Author Response
Dear Reviewer, we have attached a file with the answers to your comments.

Reviewer 4 Report
Comments and Suggestions for Authors
After another evaluation of the manuscript, I see some improvement in the quality of the paper. The authors have accepted some of my requests and there are still adjustments to be made:
L.2- "Fortification " - I was wondering if you are following the WHO in considering and using the term "fortification"? Or some resolution from the authors' country? I think it would be important to review the use of the term. There are some basic requirements.
L.81- sensorial, should be sensory
L. 82- characteristics, should be attributes
L. 83 - smell or odor ?
L.130 and 131- "percentages was chosen based both on literature
reports regarding either the same fruit or similar ones " - It seems to me that it would be better to quote the authors you have followed.
L.161-163 and table 1 - The place reader will look for continued without the details of the ingredients that were requested in the first version. please consider the information in the paragraph and table. - "Eg. warm water ? chemical yeast ? demerara sugar ?" salt ? refined salt ? fleur de sel ? Himalayan salt.
L.174- There are many types of bread: flatbread, French bread, focaccia, rye, pita, baguette, boule. This information is super important and should be more emphasized in the paper, perhaps even in the keywords.
L.240 - I still think that figure 3 as it stands is neither of good quality nor didactic, but it would have the potential for a better presentation.
L.519- in the previous version, the paper was mixed with these different noums ""cohesiveness, elasticity, gumminess, chewiness and resilience" X table 4 - Elasticity Resilience,
Gummosity (N) Masticability , (N) Cohesiveness" - it seems to me that you have chosen for this version the least used noums in rheology. For example, Masticability. Check other papers.
L.424- Odour - Standardize all your paper. Smell or odor ? American English is also mixed with British English.
L.429- Authors, be clearer- From 1st version - "Has the project been submitted to an evaluation by a university ethics committee? Did it follow the Helsinki
declaration? Please, enter the approval protocol number. "
L.520 - I couldn't find the firmness result, which could be in table 4.
L.654- Sensorial again
L.695- I'm sorry, but I don't think it's a good strategy to end a paper with a Figure before the conclusion.
Regarding the figure, I found it too colorful, and others blurred in gray, which ended up affecting understanding of what really needs to be highlighted.
I think the findings on PC1 and PC2 should be mentioned in the results and I suggest using a cleaner PCA format.
Comments on the Quality of English Language
The English could be improved to more clearly express the research.
Many words were not chosen in technical English.
English is always useful to ask a native speaker for a final appreciation.
Author Response

(The authors gave the same response as above.)
